# A new roughness length parameterization accounting for wind-wave (mis)alignment

Sara Porchetta[1 & 2], Orkun Temel[2], Domingo Muñoz-Esparza[3], Joachim Reuder[4], Jaak Monbaliu[5], Jeroen van Beeck[2], and Nicole van Lipzig[1]

[1]KULeuven, Department Earth and Environmental Sciences, Leuven, Belgium
[2]von Karman Institute for Fluid Dynamics, Sint-Genesius-Rode, Belgium
[3]National Center for Atmospheric Research, Boulder, Colorado, USA
[4]Geophysical Institute, University of Bergen, and Bjerknes Centre for Climate Research, Bergen, Norway
[5]KULeuven, Department of Civil Engineering, Leuven, Belgium

**Correspondence:** Sara Porchetta (sara.porchetta@vki.ac.be)

**Abstract.** Two-way feedback occurs between offshore wind and waves. However, the influence of the waves on the wind profile remains understudied, in particular the momentum transfer between the sea surface and the atmosphere. Previous studies showed that for swell waves it is possible to have increasing wind speeds in case of aligned wind-wave directions. However, the opposite is valid for opposed wind-wave directions, where a decrease in wind velocity is observed. Up to now, this behavior has not been included in most numerical models due to the lack of an appropriate parameterization of the resulting effective roughness length. Using an extensive data set of offshore measurements in the North Sea and the Atlantic Ocean, we show that the wave roughness length affecting the wind is indeed dependent on the alignment between the wind and wave direction. Moreover, we propose a new roughness length parameterization taking into account the dependence on alignment, consisting of an enhanced roughness length for increasing misalignment. Using this new roughness length parameterization in numerical models might facilitate a better representation of offshore wind, which is relevant to many applications including offshore wind energy and climate modeling.

*Copyright statement.* TEXT

## 1 Introduction

During the past years there has been an increased interest in wind turbines. Wind energy has been proposed as an ideal alternative for non-clean energy sources, and a good candidate to meet the rising energy demands. Moreover, there is more and more interest in offshore wind turbines because of their societal benefits and their higher wind extraction possibilities compared to onshore wind turbines. The last ten years the European offshore capacity increased by 11 GW (WindEurope Business Intelligence, 2017). Additionally, by 2020, 20% of the total energy should be renewable, this in order to meet the renewable energy directive (Directive 2009/28/EC). As such the estimated installed wind energy capacity by then will be 40 GW (EWEA, 2011). How-

ever, due to the high cost of offshore wind turbines, it is important to have accurate information about the vertical structure of the wind profile at offshore wind farm locations. Such accurate profiles will help estimate the energy production, the dynamic loads and fatigue, which influence the design of the wind turbine, the operation of the wind farm, the wind turbine wakes and finally the lay-out and allocation of new wind farms. In order to model offshore wind profiles accurately, the wind-wave interaction should be better understood. This physical understanding will result in more physical relationships that can be included in coupled atmosphere-wave models. Hence, it is important to have offshore measurement data available to improve our understanding of the wind-wave interaction.

The wind-wave interaction mainly occurs within the lowest part of the Marine Atmospheric Boundary Layer (MABL), directly influenced by the sea surface. Numerical studies suggest that the impact of the waves can extent up to the wind turbine hub height, nowadays typically 100 m (Sullivan et al., 2008; Patton et al., 2015; Nilsson et al. , 2012). Apart from the wind-wave interaction studied here, there are also other factors affecting the MABL like the sea surface temperature, sea spray, breaking waves etc. Contrary to the atmospheric boundary layer (ABL) over land, the diurnal cycle of the atmospheric stability offshore is negligible due to the high heat capacity of the ocean. Moreover, the characteristics of the MABL are mainly influenced by the variations in the momentum flux from the sea surface to the atmosphere, such as varying wave length, wave speed and wave height. In addition, higher wind speeds with lower turbulence intensities are present for MABLs, which is related to a reduced roughness of the ocean compared to over land (Ardhuin et al., 2009; Stull, 1988). Throughout the years, the wind-wave interaction has been thoroughly investigated, both by numerical and experimental characterization (Sullivan et al., 2008; Kalvig et al., 2013; Drennan et al., 2005; Edson et al., 2007; Li et al., 2018). However, the effect of alignment between wind and wave directions has not received sufficient attention, while there are preliminary indications that it can have an impact on momentum transfer (Grachev et al., 2003; Patton et al., 2015; Drennan et al., 2005).

The goal of this paper is threefold. First we would like to identify from observations if the roughness length is dependent on the alignment between the wind and peak wave direction. Second, we aim at developing a method to include this alignment effect in atmospheric models. Third, we apply the method mentioned before to derive a specific parameterization for the atmospheric models based on the limited existing observations. The parameterization can subsequently be improved when more data become available.

## 2    State of the art in parameterizing wind-wave interactions

In order to investigate wind-wave interactions, a good understanding of the momentum transfer between the sea surface and the atmosphere is necessary. Critical in modeling these complex interactions (Fig. 1) is the roughness length parameterization.

An important way through which the momentum transfer is correctly represented in numerical models is by imposing the right shear stress, which depends on the roughness length parameterization. The total shear stress, $\tau_{tot}$, and thus the offshore

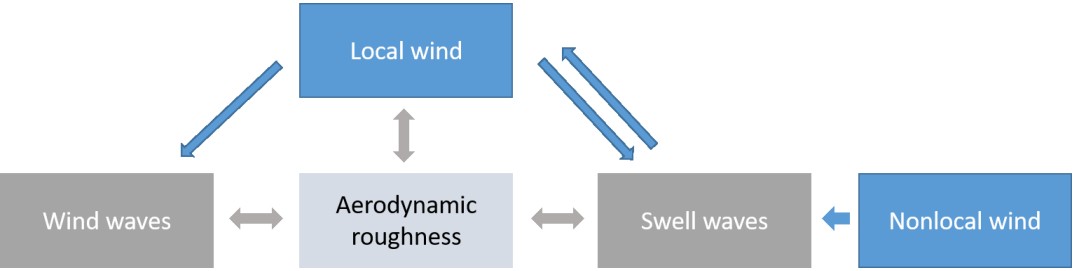

**Figure 1.** Schematic of the wind-wave interactions, upward momentum transfer to the atmosphere or negative shear is possible in case of swell waves. Swell waves are generated by nonlocal wind and can travel long distances. Local winds further influence swell waves, but can also produce wind waves through downward momentum transfer. In atmospheric models, a parameterization of the aerodynamic roughness length is used to represent all these processes.

momentum transfer, is dependent on the turbulent shear stress, $\tau_{turb}$, the wave induced shear stress, $\tau_{wave}$, and the viscous shear stress, $\tau_{visc}$ (Phillips, 1977).

$$\tau_{tot}(z) = \tau_{turb}(z) + \tau_{wave}(z) + \tau_{visc}(z) \tag{1}$$

The viscous shear stress is assumed to be negligible because of the MABL being characterized by a high Reynolds number. The wave shear stress, however, is considerable in the MABL and its effect decreases with height. As can be seen in Fig. 1, momentum transfer can go two ways; from the wind to the waves and vice versa. The sign of the wave shear stress is strongly dependent on the wave age, which is most often defined as

$$\chi = \frac{c_p}{U_{10}} \tag{2}$$

In this equation $U_{10}$ is the wind speed at 10 m and $c_p$ is the wave phase speed at the peak of the wave energy spectrum, hereafter referred to as wave speed. When $\chi < 1.2$, the wind is dominant and the waves present are called wind waves or young waves. As $\chi$ approaches 1.2, wind and waves reach equilibruim and the sea is fully developed. However, when $\chi > 1.2$, the wave speed is dominant over the wind speed and we speak about swell waves or old waves (Donelan, 2011). This can also be written as a function of the air friction velocity, $u_*$, shown in Eq. (3) where $\chi < 20$ corresponds to young waves and $\chi > 20$ to old waves (Drennan et al., 2003).

$$\chi = \frac{c_p}{u_*} \tag{3}$$

For swell waves the wave shear stress, and thus the momentum flux caused by it, is small compared to the total shear stress. However, this is not the case for young waves, where most of the momentum flux is determined by the wave stress. As such,

it is important to include the wave age parameter in a roughness length parameterization (Janssen, 1991), as will be shown later. For wind waves, the wave shear stress is positive. As such, there is a downward momentum flux from the atmosphere to the sea surface, and the aerodynamic roughness length ($z_0$) increases with increasing shear stress. On the other hand, for swell waves, the aerodynamic roughness length can decrease down to a point where the wave shear stress can become negative. This

can cause the total shear stress to become negative which results in a upward momentum flux. Because of this, momentum is transported from the sea surface to the atmosphere (Cathelain, 2017; Sullivan et al., 2000).

In order to model the momentum transfer from the atmosphere to the sea correctly, most atmospheric models use the Charnock aerodynamical roughness length parameterization (Eq. (4)), where $\alpha$ is the Charnock parameter that depends on the sea state and is assumed to be constant and $g$ is the gravitational constant (Charnock, 1955).

$$z_0 = \alpha \frac{u_*^2}{g} \tag{4}$$

An important application where this parameterization is used is in atmospheric models uncoupled to an ocean model, an example of which is the Weather Research and Forecasting (WRF) model (Skamarock et al., 2008), where the Charnock parameter used for offshore conditions is constant and equal to 0.018. In this model the Charnock constant is only derived for fully developed, wind waves over deep water. Jiménez and Dudhia (2018) recommend to have a modified sea surface roughness

length formulation for shallow waters as this formulation for deep water is resulting in a positive wind speed bias. Additionally, Larsén et al. (2012) showed that the wind at hub height is underestimated for storm conditions, caused by this simple roughness length parameterization proposed by Charnock (1955).

Hsu (1973) suggested that the Charnock parameter should include information of the sea surface characteristics. As such, it included the wave steepness $H_s/L$ implicitly, where $H_s$ is the significant wave height and $L$ the wave length of the dominant

waves. For deep water waves, where the depth of the sea (h) is bigger than half the wave length, the wave phase speed is related to the wave length by Eq. (5), while for shallow regions the wave phase speed it is equal to Eq. (6). For the calculation of the wave phase speed the full dispersion relation is used in this study.

$$c_p = \sqrt{\frac{gL}{2\pi}} \tag{5}$$

$$c_p = \sqrt{gh} \tag{6}$$

Hsu (1973) modified the roughness length parameterization based on the available field and laboratory measurements for near-neutral stability and deep water, which resulted in Eq. (7), where $A$ is a constant.

$$\frac{z_0}{H_s} = A \left( \frac{u_*}{c_p} \right)^2 \tag{7}$$

A more general roughness length parameterization was found by Donelan (1990) and is shown in Eq. (8).

$$\frac{z_0}{H_s} = A \left( \frac{u_*}{c_p} \right)^B \tag{8}$$

However, no consensus for the constants $A$ and $B$ was found because every data set resulted in different values. As such, Drennan et al. (2003) tried to avoid this problem by estimating the roughness length relation using multiple data, taking into account a wide range of variable offshore conditions. Drennan et al. (2003) found Eq. (9) to be an improved roughness length parameterization, especially for pure wind-sea, rough-flow, deep-water data. The parameterization was used by Bruneau and Toumi (2016) for the study of a fully coupled atmosphere-ocean-wave model for the Caspian Sea.

$$\frac{z_0}{H_s} = 3.35 \left( \frac{u_*}{c_p} \right)^{3.4} \tag{9}$$

Almost simultaneously with Donelan (1990), Maat et al. (1991) suggested that the Charnock parameter should be a function of the wave age and proposed Eq. (10) for the Charnock parameter.

$$\alpha = \mu \left( \frac{c_p}{u_*} \right)^n \tag{10}$$

The parameters $\mu$ and $n$ are equal to 0.8 and -1 respectively and are obtained from measurements of the HEXOS campaign 9 km out of the Dutch coast (Katsaros at al., 1987). Based on the same measurements, Smith et al. (1992) found $\mu$ equal to 0.43 and $n$ equal to -0.96. Even though both authors have different parameters, they agree that young waves are rougher than older ones. Other values for $\mu$ and $n$ have been proposed by Monbaliu (1994), which used the HEXOS campaign and Vickers and Mahrt (1997) and Johnson et al. (1998), both using the RASEX campaign. Clearly, different sites result in different constants for the roughness length parameterization. The approach of Drennan et al. (2003) combines different data sets from different measurement sites and obtains only one set of constants.

An alternative roughness length parameterization was proposed by Taylor and Yelland (2001), which is based on the wave steepness, Eq. (11), where $L_p$ is the wave length at the peak wave spectrum.

$$\frac{z_0}{H_s} = 1200 \left( \frac{H_s}{L_p} \right)^{4.5} \tag{11}$$

To obtain this roughness length parameterization three measurement data sets describing different sea states were used. This parameterization is used in coupled atmosphere wave models by Warner et al. (2010) and Bolaños et al. (2014). Warner et al. (2010) used the new roughness length parameterization by Taylor and Yelland (2001) in the Coupled Atmosphere Ocean Wave Sediment Transport (COAWST) model. This model couples the WRF atmospheric model with the SWAN wave model by using Eq. (11) as a boundary condition for the atmospheric model. To enable this, the wave length and wave height obtained by the wave model are passed on to the atmospheric model. Bolaños et al. (2014) used the roughness length parameterization

shown in Eq. (11) but did not actively couple it to a wave model. Instead, the wave length and height of the roughness length parameterization are estimated based on the $10\ m$ wind speed assuming fully developed sea conditions.

Drennan et al. (2005) compared the performance of the roughness length parameterization of Taylor and Yelland (2001) and Drennan et al. (2003) on eight distinct data sets corresponding to different sea states. This comparison resulted in a good
performance of the roughness length parameterizations for young waves, especially for the measurement data that were used to develop the parameterizations. However, these roughness length parameterization performed poorly in regions of swell and Drennan et al. (2005) suggested that a more elaborated roughness length parameterization including not only the swell magnitude but also the direction of the swell waves could improve the model. The effect of the stress vector not aligned with the main wind, but somewhere between the wind direction and the wave direction was first described by Rieder et al. (1994).
Additional roughness length parameterizations were proposed by Janssen (1991), Fan et al. (2012) and Liu et al. (2011), all with a different focus (e.g., the later including a sea-spray induced roughness length).

The focus of this paper, however, will be on the influence of the difference between the peak wave direction, hereafter referred to as wave direction, and the wind direction on the roughness length parameterization. While a wind-wave direction-based roughness length parameterization has not yet been investigated, the importance of this effect has been suggested by
Drennan et al. (2005) based on experimental observations. The importance of the swell direction has also been investigated by numerical simulations. Sullivan et al. (2008) performed Large Eddy Simulations (LES) and found that the drag of aligned swell waves is much smaller than that of opposed swell waves, where the latter represents waves with phase speeds opposite to the wind direction. Likewise, Kalvig et al. (2013) looked into the wave wind alignment applying the Reynolds-Averaging Navier-Stokes (RANS) equations and found that swell waves opposed to the wind can ensure reduced and even reversed wind speeds
in the lowest meters. LES investigations by Patton et al. (2015) found that the wind speed at hub-height of wind turbines can decrease by 15% for opposed wind-wave alignment compared to aligned cases, with turbulence intensities increased by a factor two for opposed cases. All these numerical modeling efforts indicate that the correct representation of the momentum fluxes between the sea surface and the atmosphere, and thus a good roughness length parameterization, are of crucial importance. An attempt for a new roughness length parameterization was made by Patton et al. (2015) based on their LES results, including
the effect of the alignment between the wind and waves. This parameterization is based on Drennan et al. (2003) and shown in Eq. (12), where $\theta$ is the angle between the wave direction and the wind direction.

$$\frac{z_0}{H_s} = A \left( \frac{u_*}{c_p} \right)^{3.4cos\theta} \tag{12}$$

Unfortunately, this new roughness length parameterization is not finalized yet. For young wave ages, unrealistic dimensionless roughness length values are obtained and the constant $A$ is undefined. Additionally, this parameterization has only been
tested on the results of LES simulations. Lastly, these simulations included only imposed waves and a one-way wind-wave interaction. As such, only the effects of the waves on the wind, were studied. Important here is that Drennan et al. (2003) and Patton et al. (2015) suggested that a new roughness length parameterization should include the angle between the wind and the wave direction, consistent with the wind profiles obtained by the LES and RANS simulations of Sullivan et al. (2000) and

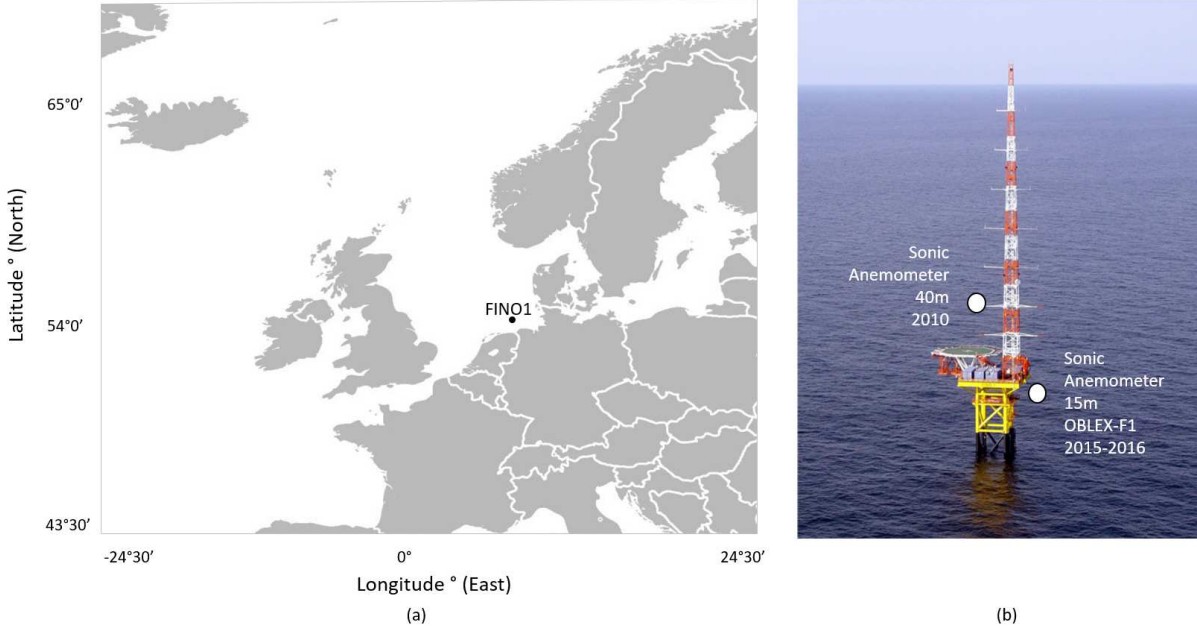

**Figure 2.** (a) Location of the FINO1 measurement mast in the North Sea. (b) View of the FINO1 measurement mast and the location of the two sonic anemometers used to obtain the new roughness length parameterization (Modified from Muñoz-Esparza et al. (2012)).

Kalvig et al. (2013). In this study we investigate the presence, and impact, of a possible effect of the wind-wave alignment using offshore measurements. With these results we propose a new roughness length parameterization, to support future research on the improvement of numerical modeling of the transfer of momentum between the sea surface and the atmosphere, in particular for opposed wind-wave directions.

## 3   Methods

### 3.1   Offshore field measurement data sets

The *Forschungsplattformen in Nord- und Ostsee Nr.1* (FINO1) is one of the few offshore measurement platforms where measurements are simultaneously obtained for both atmospheric and oceanographic parameters. The measurement mast is located in the North Sea, 45 km north of the coast of Borkum Island, Germany. The exact coordinates of the mast are N 54°0'53.5" E 6°35'15.5", as shown on Fig. 2 (A). The measurement tower is exposed to an unlimited fetch area for northwesterly to north wind directions, while in the other directions it is fetch limited due to coastal presence of the Netherlands, Germany and Denmark. The measurement mast itself extends 100 m above mean sea level and the mean water depth at this specific location is 30 m. Measurements are continuously collected since 2003.

The measurement mast is equipped with different sensors to measure wind speed and direction, air and sea temperature, air pressure, and humidity. Cup anemometers measure the velocity at 33, 40, 50, 60, 70, 80, 90 and 100 m while sonic anemometers with a frequency of 10 Hz are only present at 40, 60 and 80 m (Fig. 2 (B)), with an orientation of 308°. In order to eliminate the wind shadow zone caused by the measurement mast, the wind directions between 60° and 200° are removed. During

the OBLEX-F1 campaign led by Christian Michelsen Researh (CMR) and Universitetet i Bergen (UiB) two additional sonic anemometers where installed at 15 and 20 m above mean sea level, with a measurement frequency of 25 Hz and an orientation of 135°. The wind shadow zone is thus between 245° and 360° and excluded from the further analyses. Especially the anemometer at 15 m provides critical information about the wind-wave interaction (Fig. 2 (B)). Additionally, low response (1 Hz) pressure measurements were taken at 20 m, humidity measurements at 30, 50 and 90 m, air temperature at 40, 50, 70 and 100 m and

precipitation measurements at 20 m and 90 m. Wave information, including significant wave height, peak wave direction and (peak) wave period are measured by a Datawell/MKIII buoy in the close vicinity of the FINO1 measurement mast. For this study two main measurement campaigns were found to contain momentum and heat flux measurements, which are essential for the derivation of the new roughness length parameterization. One held from January to December 2010 (Muñoz-Esparza, 2013; Beeken et al., 2008), including the necessary flux information at 40 m altitude while during the OBFLEX-F1 campaign

flux measurements were taken from May 2015 to September 2016 at an altitude of 15 m.

The *Air-Sea Interaction Tower* (ASIT) is located 3.2 km south of Martha's Vineyard (N 41° 19.5', W 70° 34.0') in the Atlantic Ocean (Fig. 3 (A) & (B), point C) and is part of the CBLAST measurement campaign (Edson et al., 2007; Chen et al., 2007). At the location of the ASIT tower the water is around 15 m deep. Measurements are continuously available between 2003 and 2012. Wind coming from directions between 140° and 250° are fetch unlimited, as it is not influenced by the land to

the north nor by shallow water to the east and west. Moreover, directions between 0° and 150° are affected by flow distortions by the tower itself (Edson et al., 2007), resulting in wind shadow distortions. The ASIT tower is designed as a low-profile, fixed structure in order to minimize these flow distortions. Even though in theory no correction is required, it was decided to exclude measurements from the wind shadow region to ensure no wind shadow effects and for consistency in methodology.

The instrumentation of the ASIT tower is shown on Fig. 3 (C). In this study on wind- wave interaction, measurements from

a Gill R3-sonic anemometer are used. These measurements are taken 18 m above sea level with a measuring frequency of 20 Hz. Fluxes of momentum and virtual heat are calculated for 20 minute averaging periods. Slow response measurements of air temperature, pressure and relative humidity are available from a vaisala RH/T sensor. The oceanic variables are obtained from the subsurface node (Fig. 3 (B), point B) located 1.5 km south of Martha's Vineyard at 12 m depth. Wave characteristics, such as significant wave height and peak wave direction are measured here. Even though the oceanographic data and the

atmospheric data are not co-located, we assume that the marginal distance of 1.7 km and a water depth difference of 3 m are small enough to assume the same oceanographic features. Furthermore, the wind measurements are averaged over 20 minutes, a time period during which the wave conditions are not expected to change significantly. When comparing the slow response atmospheric measurements (wind speed, wind direction, temperature ...) between the meteorological mast (Fig. 3 (B), point A) and the ASIT tower (Fig. 3 (B), point C), no significant differences have been noticed. For these reasons we assume that the

ASIT measurements can be combined with the subsurface node to investigate the wave induced velocity changes. The same

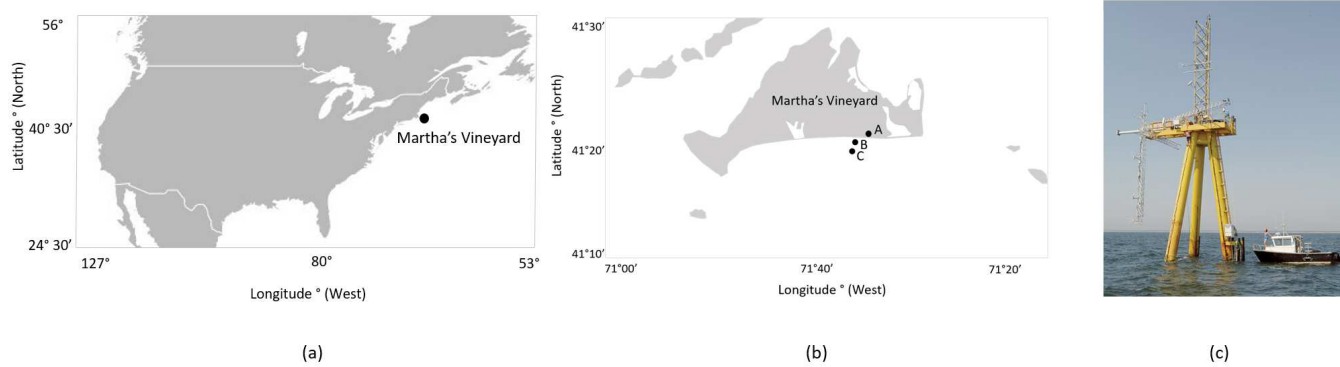

(a)                                          (b)                                      (c)

**Figure 3.** (a) Location of the ASIT measurement mast in the Atlantic Ocean. (b) Location of the ASIT measurement mast compared to Martha's Vineyard, point A is the meteorological mast, point B is a subsurface node and point C is the ASIT measurement tower (c) ASIT measurement tower.

assumption was also made by Sullivan et al. (2008). A summary of which data is used for the two different measurement masts is shown in Table 1.

**Table 1.** Summary of measurement data used from ASIT and FINO1.

| Location | Year | Wind shadow zone | Altitude $[m]$ | Water depth $[m]$ |
|----------|------|------------------|----------------|-------------------|
| FINO1 | 2010 | 60°-200° | 40 | 30 |
| FINO1 | 2015-2016 | 245°-360° | 15 | 30 |
| ASIT | 2003-2012 | 0°-150° | 18 | 15 |

### 3.2 Data selection and processing

From the 2003-2012 period, the measurements of 2005, 2009 and 2012 of the ASIT mast were excluded, because they did not
5   contain high resolution velocity and temperature measurements coupled with the simultaneously occurring wave parameters. No such exclusions were necessary for the FINO1 data. Additionally, the wind shadow zones resulting in flow distortion were excluded for both data sets. Moreover, wind speeds below 1 m/s were removed from the data sets, because in these conditions the uncertainty on the mean wind direction increases (Anfossi et al., 2005). These corrections resulted in a total of 74 108 measurements from both measurement masts. Combining observations from different locations, similar to the approach by
10   Drennan et al. (2003), reduces the effect of site specific parameters and a more general form can be found.

    An investigation of the dominant wind and wave direction for the two measurement locations is presented in Appendix A. In this study the effect of the (mis)alignment between the wind and wave direction is of major interest. The histogram of the

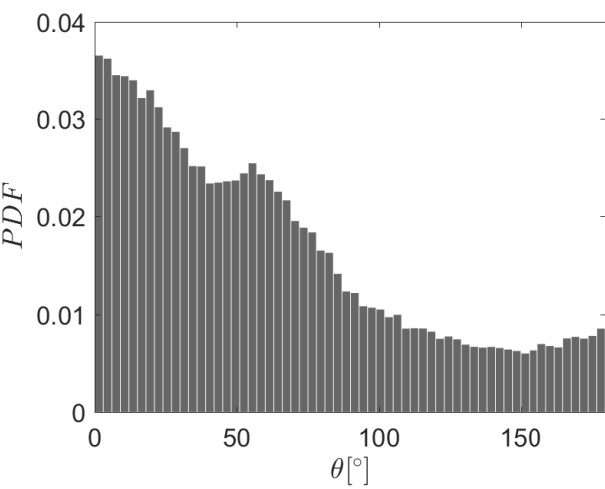

**Figure 4.** Probability density function (PDF) of the angle between the wind and peak wave direction for both ASIT and FINO measurements.

angle between the wind and peak wave direction is shown in Fig. 4. Wind-wave direction alignment ($\theta < 30°$) occurs most of the time (33%), however, as can be seen from Fig. 4, there is significant probability of occurrence of misalignment events of different degrees, while opposed wind and wave directions ($\theta > 150°$) is a less frequent scenario (8%). This behavior is seen for both FINO1 and ASIT measurements. The difference between ASIT and FINO1 is that for the ASIT measurement mast 90° misalignments are more present than for the FINO1 measurements, this could be due to more fetch unlimited areas at ASIT.

In order to validate the ASIT and FINO1 measurements against the existing roughness length parameterization of Drennan et al. (2003), we processed the data following Drennan et al. (2005). The choice of the parameterization of Drennan et al. (2003) as a starting point comes from the fact that Patton et al. (2015) already set the first step in improving this roughness length parameterization based on their simulations. Moreover, the law of Drennan et al. (2003) is already implemented in recent models (Bruneau and Toumi, 2016) and can be easily incorporated in other coupled wave-atmosphere models (Warner et al., 2010).

A first step required is the calculation of the friction velocity (Eq. (13)), which is based on the measured alongwind, $< w'u' >$, and crosswind, $< w'v' >$ kinematic momentum fluxes (Phillips, 1977). For the ASIT measurement tower these velocity flux measurements where calculated using the eddy correlation technique with an averaging period of 20 minutes and were already available. For the FINO1 measurement campaign of 2010 the flux measurements where processed by Muñoz-Esparza et al. (2012), also using the eddy correlation technique and averaging over 30 minute periods, and with the averaging time determined from an Ogive analysis. For the additional FINO1 measurement campaign of 2015-2016 only the raw data were available. The fluxes for this data set were calculated using the eddy correlation technique implemented in

EddyPro© (v 6.2.1; standard settings) and averaged every 30 minutes, consistently with the FINO1 2010 data set. This software is extensively used in atmospheric sciences (Mammarella et al., 2016; Fratini and Mauder, 2014).

$$u_*^2 = \left[ (- <w'u'>)^2 + (- <w'v'>)^2 \right]^{0.5} \tag{13}$$

The aerodynamic roughness length, $z_0$, is calculated from the logarithmic wind profile assumption shown in Eq. (14), where $\kappa$ is the von Karman constant, $U_z$ is the corrected wind speed and $U_0$ is the surface drift speed. The latter is small and assumed to be zero.

$$\frac{1}{\kappa} log \frac{z}{z_0} = \frac{U_z - U_0}{u_*} \tag{14}$$

The corrected wind speed, $U_z$, is calculated based on Eq.(15), where $U_{z0}$ is the wind speed at a height of $z$ m above sea level and $\psi_u$ is the integrated stability function according to Barthelmie (1999).

$$U_z = U_{z0} + \frac{u_*}{\kappa} \psi_u \left( \frac{z}{L} \right) \tag{15}$$

In order to find the integrated stability function, $\psi_u$, the stability of the atmosphere should be classified in stable and unstable atmospheric conditions. This distinction is made based on the Obukhov length (Eq. (16)), where $\theta_v$ is the virtual potential temperature and $<w'\theta_v'>$ is the virtual potential temperature flux (Donelan, 1990). For the ASIT measurements, the latter is directly available from the measurements and averaged over 20 min intervals while the former is calculated based on the Clausius-Clapeyron relation, employing the relative humidity and the pressure. For the FINO1 measurements both the virtual potential temperature as well as the virtual potential temperature flux were computed by Muñoz-Esparza et al. (2012) or directly available as output from the EddyPro© software.

$$L = -\frac{u_*^3 \theta_v}{g\kappa <w'\theta_v'>} \tag{16}$$

The classification of the atmosphere in different stability classes according to Wijk et al. (1990) is shown in Table 2. Both very stable and unstable atmospheric conditions occur more frequently, which is consistent with literature. For the FINO1 measurement mast unstable conditions are often present due to cold air advecting above the much warmer ocean, this is consistent with other locations in the North-Sea (Patton et al., 2015; Barthelmie, 1999). However, for the ASIT measurement mast stable conditions do occur often in late spring to early summer when the ocean is slowly warming by warm ocean water flowing over the colder ocean, typically identified with cool summer weather and fog (Edson et al., 2007; Crofoot, 2004). The integrated stability function can be calculated by Eq. (17) for stable conditions (L > 0) and by a combination of Eq. (18) and Eq. (19) for unstable conditions (L < 0) (Barthelmie, 1999).

**Table 2.** Stability classification based on Obukhov Length, $L$, for the combined ASIT and FINO1 data set

| Stability class | Range of L | Frequency |
|---|---|---|
| Very Stable | $0 \leq L < 200$ | 36.58 % |
| Stable | $200 \leq L < 1000$ | 7.16 % |
| Near-Neutral | $1000 \leq |L|$ | 4.19 % |
| Unstable | $-1000 < L \leq -200$ | 9.56 % |
| Very Unstable | $-200 < L \leq 0$ | 42.52 % |

$$\psi = -5\frac{z}{L} \tag{17}$$

$$\psi = 2ln\left(\frac{1+x}{2}\right) + ln\left(\frac{1+x^2}{2}\right) - 2tan^{-1}x + \frac{\pi}{2} \tag{18}$$

$$x = \left[1 - 16\left(\frac{z}{L}\right)\right]^{0.25} \tag{19}$$

## 4  Results and discussion

To validate the roughness length parameterization of Drennan et al. (2003), the dimensionless roughness length, $z_0/H_s$, is
plotted against the inverse wave age parameter $u_*/c_p$ (Fig. 5). The curve proposed by Drennan et al. (2003) performs well
in regions of young waves, however, for swell dominated seas (low $u_*/c_p$) the proposed roughness length parameterization
performs poorly. This indicates that the roughness length of swell waves is not only dependent on the wave age, but also on
other parameters, like the difference between the wave direction and wind direction. The same validation can be done for
the Charnock roughness length parameterization, used in many numerical mesoscale codes, by plotting the roughness length
against the friction velocity. However, multiple studies like Drennan et al. (2003),Taylor and Yelland (2001), Janssen (1991),
Fan et al. (2012) and Liu et al. (2011) among many others recognized to include more information about the sea state as this
improves the estimation of the roughness length. As these studies were a starting point to improve the Charnock's roughness
length parameterization for numerical models by including extra information from the sea state, our goal is to build further
upon this by taking into account the alignment between the wind and wave direction.

While Drennan et al. (2003) improved the roughness length parameterization by including wave age and taking into account
multiple measurement sites, one major drawback remained. As the (mis)alignment between wind and waves was not consid-
ered, even though research has shown that this might be important (Grachev et al., 2003; Drennan et al., 2003; Sullivan et al.,

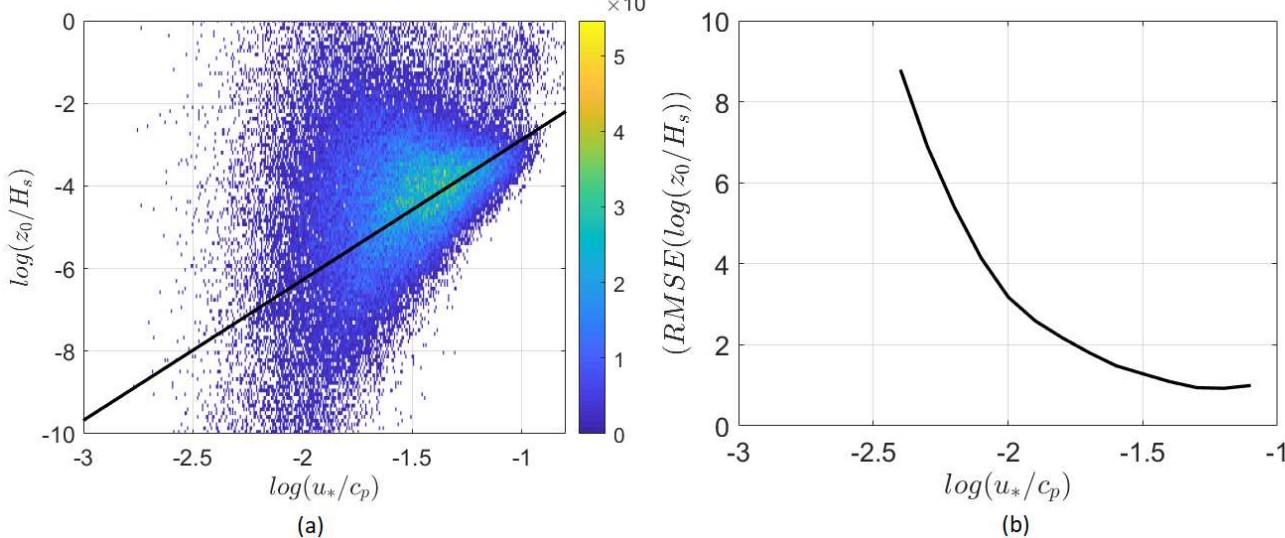

**Figure 5.** (a) The dimensionless roughness length, $z_0/H_s$, plotted against the inverse wave age parameter $u_*/c_p$ for the combined data set of 74 108 points (ASIT and FINO1). The solid line represents the roughness length parameterization proposed by Drennan et al. (2003), Eq. (9).The color scale to the right indicates the probability of occurrence (%) of the measurement points. (b) The root mean square error of the measurement points compared to the parameterization proposed by Drennan et al. (2003).

.

2008; Kalvig et al., 2013; Patton et al., 2015). In order to see if there is an effect of the (mis)alignment of the wind-wave direction, the dimensionless roughness length is divided into six groups based on the degree of alignment. The alignment is calculated by taking the absolute value of the difference in direction between the wave and wind propagation. The frequency of measurement points in each of these groups is shown in Table 3; 0° corresponds to waves traveling in the same direction as the wind, while 180° corresponds to waves with an opposed direction to the wind direction. Comparing the probability density function of the dimensionless roughness length of these six groups (Fig. 6), it is found that the dimensionless roughness length increases for increasing misalignment. This finding is confirmed by an analysis of variance statistical test where we tested if there were differences between the group means. All alignment groups showed that they significantly (p-value < 0.05) differ from each other, except the group of 120°-150° which did not show a significantly different (p-value > 0.05) behavior to the group of 150°-180°.

Previous studies not only predicted that roughness length depends on the degree of (mis)alignment, they also implied that the effect of a reduced roughness length is more pronounced for swell waves aligned with the wind compared to young waves aligned with the wind, due to a reduced shear stress. This reduction in shear stress should thus be more pronounced at higher wave ages. To investigate this effect, the inverse wave age parameter is subdivided in three bins, bin 1 corresponds to $2 \cdot 10^{-2.75} < u_*/c_p \leq 2 \cdot 10^{-2.15}$, bin 2 to $2 \cdot 10^{-2.15} < u_*/c_p \leq 2 \cdot 10^{-1.9}$ and bin 3 to $2 \cdot 10^{-1.9} < u_*/c_p \leq 2 \cdot 10^{-1}$. These bins are obtained in such a way that the same amount of measurement points is present in each bin. The results indicate that

**Table 3.** Frequency measurement points in each alignment section and the dimensionless roughness length corresponding to the maximum probability for each section, for the combined ASIT and FINO1 data set.

| $\theta°$ | Frequency | $\log(z_0/H_s)$ |
|---|---|---|
| 0° - 30° | 32.77% | -4.7246 |
| 30° - 60° | 24.46% | -4.8478 |
| 60° - 90° | 18.35% | -4.5433 |
| 90° - 120° | 9.79% | -4.2234 |
| 120° - 150° | 7.05% | -3.9994 |
| 150° - 180° | 7 58 % | -3.9314 |

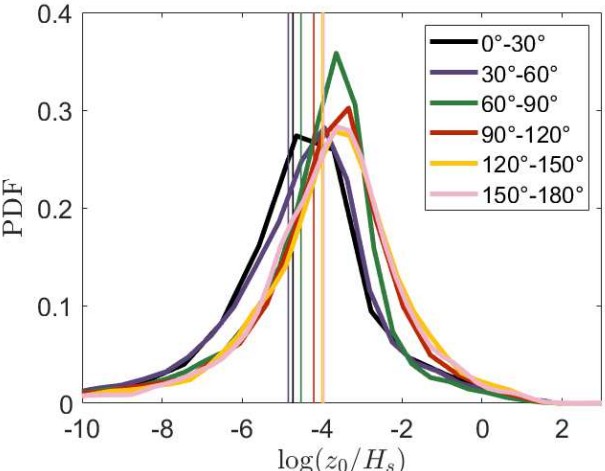

**Figure 6.** Probability density function of the dimensionless roughness length parameter for six different alignment groups, from aligned to opposed cases for the combined ASIT and FINO1 data set. The vertical lines represent the mean value of the different alignment groups.

.

the reduced roughness length for aligned wind and wave direction is more pronounced at low inverse wave age numbers (swell waves, bin 1), while for high inverse wave age numbers (wind waves, bin 3) the difference between the roughness length is less distinct (Fig. 7). This can be explained by a stronger difference in momentum flux for swell waves that are aligned or opposed, compared to conditions of young waves.

5     It can therefore be concluded that the roughness length is dependent on the alignment between the wind and the waves and more specifically that the roughness length increases with increasing misalignment. This effect is more pronounced for old waves. Therefore we propose a new roughness length parameterization formulated by correlating the dimensionless roughness length against the inverse wave age parameter. This is done for six groups of different degrees of (mis)alignment (0° - 30°,

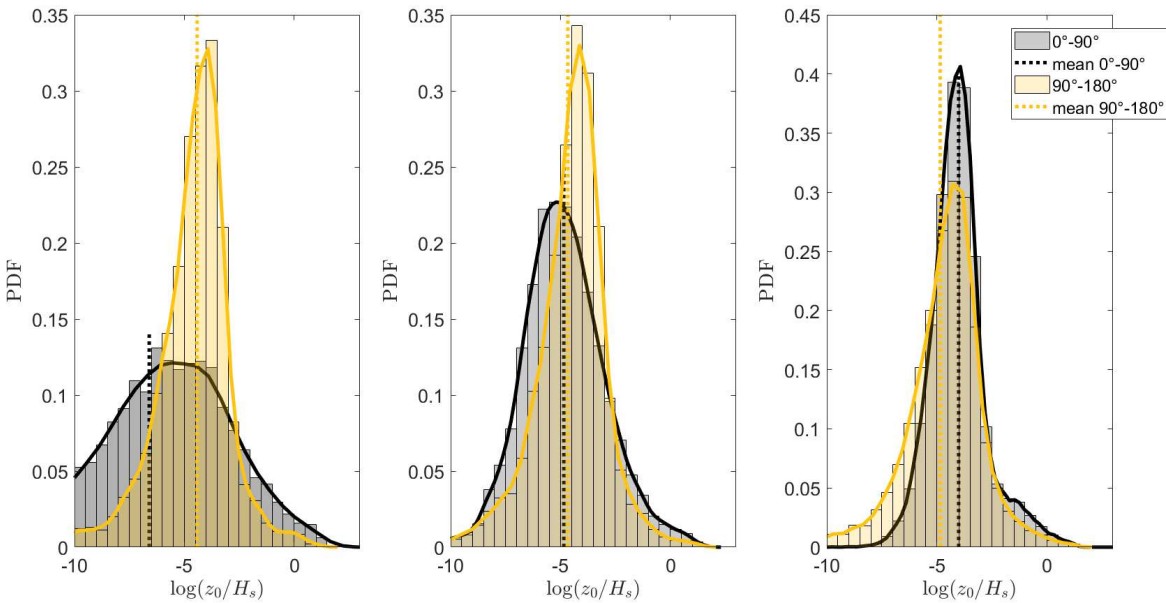

**Figure 7.** Probability density function of the dimensionless roughness length for 3 different bins of inverse wave age for both ASIT and FINO1 measurements. (a) Bin 1 corresponds to $2 \cdot 10^{-2.75} < u_*/c_p \leq 2 \cdot 10^{-2.15}$, (b) bin 2 to $2 \cdot 10^{-2.15} < u_*/c_p \leq 2 \cdot 10^{-1.9}$ and (c) bin 3 to $2 \cdot 10^{-1.9} < u_*/c_p \leq 2 \cdot 10^{-1}$. For every bin there is a difference between aligned ($\theta < 90°$) and opposed ($\theta > 90°$) wind-wave directions.

.

30° - 60°, 60° - 90°, 90° - 120°, 120° - 150°, 150° - 180°) (Fig. 8). Data was divided into inverse wave age bins ( $2 \cdot 10^{-2.5} < u_*/c_p \leq 2 \cdot 10^{-2.25}$, $2 \cdot 10^{-2.25} < u_*/c_p \leq 2 \cdot 10^{-2}$, $2 \cdot 10^{-2} < u_*/c_p \leq 2 \cdot 10^{-1.75}$, $2 \cdot 10^{-1.75} < u_*/c_p \leq 2 \cdot 10^{-1.5}$, $2 \cdot 10^{-1.5} < u_*/c_p \leq 2 \cdot 10^{-1.25}$). Bins containing less than 100 data points were excluded. This happened mostly for bin 5. The new roughness length parameterization is then developed in such a way that the mean difference of the fit of the (logarithmic)

5  bin means and the new parameterization is as small as possible. The formula for the new roughness length parameterization taking into account the alignment between the wind and the wave direction is shown in Eq. (20) and its performance is shown in Fig. 8. In this equation the angle is expressed in radians. For the exponential term a cosine function is used in agreement with Patton et al. (2015). Similarly, a cosine function is also used for the constant term for consistency between both terms.

$$\frac{z_0}{H_s} = 20cos\left(0.45\theta\right)\left(\frac{u_*}{c_p}\right)^{3.8cos(-0.32\theta)} \tag{20}$$

10  It is noted that the bathymetry and wave climates are different for the two measurement locations (FINO1 and ASIT) and that different locations yield different tuning coefficients (Maat et al., 1991; Vickers and Mahrt, 1997), and indeed the shape of the scatter plot is slightly different for FINO1 compared to ASIT. On the other hand the two sites show the same pattern with an increasing roughness length for an increasing misalignment. Moreover, the new roughness length parameterization results

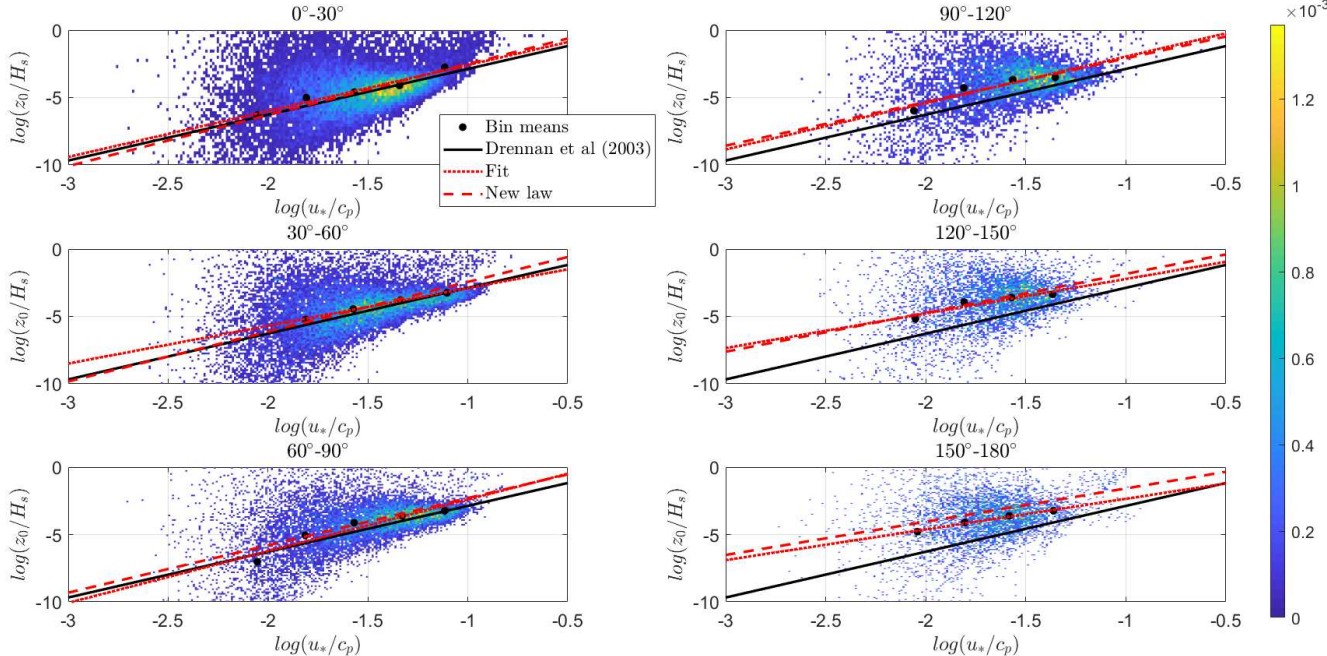

**Figure 8.** The dimensionless roughness length is plotted against the inverse wave age parameter for six different groups of alignment for the combined ASIT and FINO1 date set. In each figure the data are bin-averaged, with the bin means (logarithmic) indicated by black dots. The solid black line represents the roughness length parameterization proposed by Drennan et al. (2003), the dotted red line is the fit through the bin averages and the dashed red line is the new proposed roughness length parameterization. The color scale to the right indicates the probability of occurrence (%) of the measurement points.

in a clear improvement for opposed wind and wave directions at both sites. In the end, our parameterization has been proposed targeting implementation in mesoscale models, which can be used to simulate MABL with various wave climates. While it is possible to separate both results, the purpose of this paper was to derive a more general law, taking into account various offshore conditions. Therefore, we decided to group all data and in this way cover a range of conditions as broad as possible.

5    For a better fit at a specific location, the newly proposed parameterization can be tuned according to the data set available. Furthermore, even more different locations and conditions should be included but the availability of simultaneously measured wind and wave parameters is unfortunately scarce.

The law of Drennan et al. (2003) under-performs for different degrees of alignment, in particular for increasing degrees of misalignment between the wind and wave direction. A negative bias is even more prominent for swell waves. These results

10    show that the dimensionless roughness length is influenced by the degree of (mis)alignment. As such, they confirm that a new roughness length parameterization is justified. This is also clear in Fig. 9, where the two parameterizations are compared. The roughness length based on Drennan et al. (2003) is independent on the angle between the wind and wave direction (Fig. 9 (A)), while this is not the case for the new proposed roughness length parameterization. Here (Fig. 9 (B)) we can see, that the

roughness length, for a constant wave age, increases with increasing misalignment between the wind and wave direction. This effect is less pronounced for younger sea states ($log(u_*/c_p) > -1$), with almost no effect of the misalignment on the roughness length. Looking at the difference in roughness length prediction between the parameterization of Drennan et al. (2003) and the new roughness length including the misalignment of the direction between the wind and the waves, Fig. 9 (C), the effect of increasing roughness length is clear for increasing inverse wave age and increasing misalignment between the wind and wave direction. The measurements of FINO1 and the ASIT measurement tower, Fig. 9 (D), show that the increase of roughness length with increasing misalignment is weak in the case of swell waves. These results confirm that the new parameterization appears to be a good fit for different degree of (mis)alignment. More specifically, the slope of the curve is decreasing with increasing misalignment, indicating that more misaligned waves result in an increased roughness length. Moreover, as the difference between the new parameterization and the one from Drennan et al. (2003) is most obvious for lower inverse wave ages, the effect of misalignment on the roughness length has more impact on swell waves. This is expected, as young waves are wind generated and thus are more likely to be aligned a priori.

Note that no precipitation filter was applied on the sonic anemometer measurements, even though Zang et al. (2016) suggested to correct the sonic temperature in case of precipitation. An analysis on the FINO1 measurements of the year 2010 was done in order to investigate if the new proposed roughness length parameterization would have a systematic bias due the presence of measurements influenced by precipitation, as applying a precipitation filter to the ASIT data was not possible (Appendix B). This precipitation analyses showed no significant influence of precipitation on the new $z_0/H_s$ - $u_*/c_p$ relation.

This new roughness length parameterization, including the alignment of the wind and wave direction, reduces the scatter around the Drennan et al. (2003) parameterization considerably for misaligned cases (Table 4). The remaining scatter indicates, however, that not all relevant physical processes occurring in the MABL are adequately described by the roughness length parameterization, leaving room for future improvements. Liu et al. (2011) found that sea spray, again an interplay between wind and wave, also influences the roughness length. Furthermore, not only sea spray but also wave steepness of swell waves alter the momentum transfer between the sea and the atmosphere, which in turn influences the roughness length. The wind stress decreases if the swell steepness increases (Ocampo-Torres et al., 2011). In fact, García-Nava et al. (2012) proposed a new roughness length parameterization which includes both the effect of the wave age and the swell steepness on the roughness length. Recently, Jiménez and Dudhia (2018) also found that the roughness length parameterization should be adapted considering the depth present. Moreover, the depth should be investigated to account for wave shoaling but also to study the effect of bottom friction. As such, future work of the combined effect of wind-wave misalignment and the effect of sea spray, swell steepness and depth, are needed to further improve the roughness length parameterization for numerical models. This requires additional observational data to be taken, for example, the swell height and sea spray information were not available for our measurement sites at this moment in time Additionally, more measurement locations should be included in the analysis, in order to reduce the site specific parameters even further.

Even though, it is known that multiple parameters play an important role in the complex wind-wave interaction, in this paper we focus on one aspect namely the influence of the difference in direction between the wind and the waves. In order to exclude that the increase of roughness length with increasing misalignment is an artifact of the choice of the roughness parameteriza-

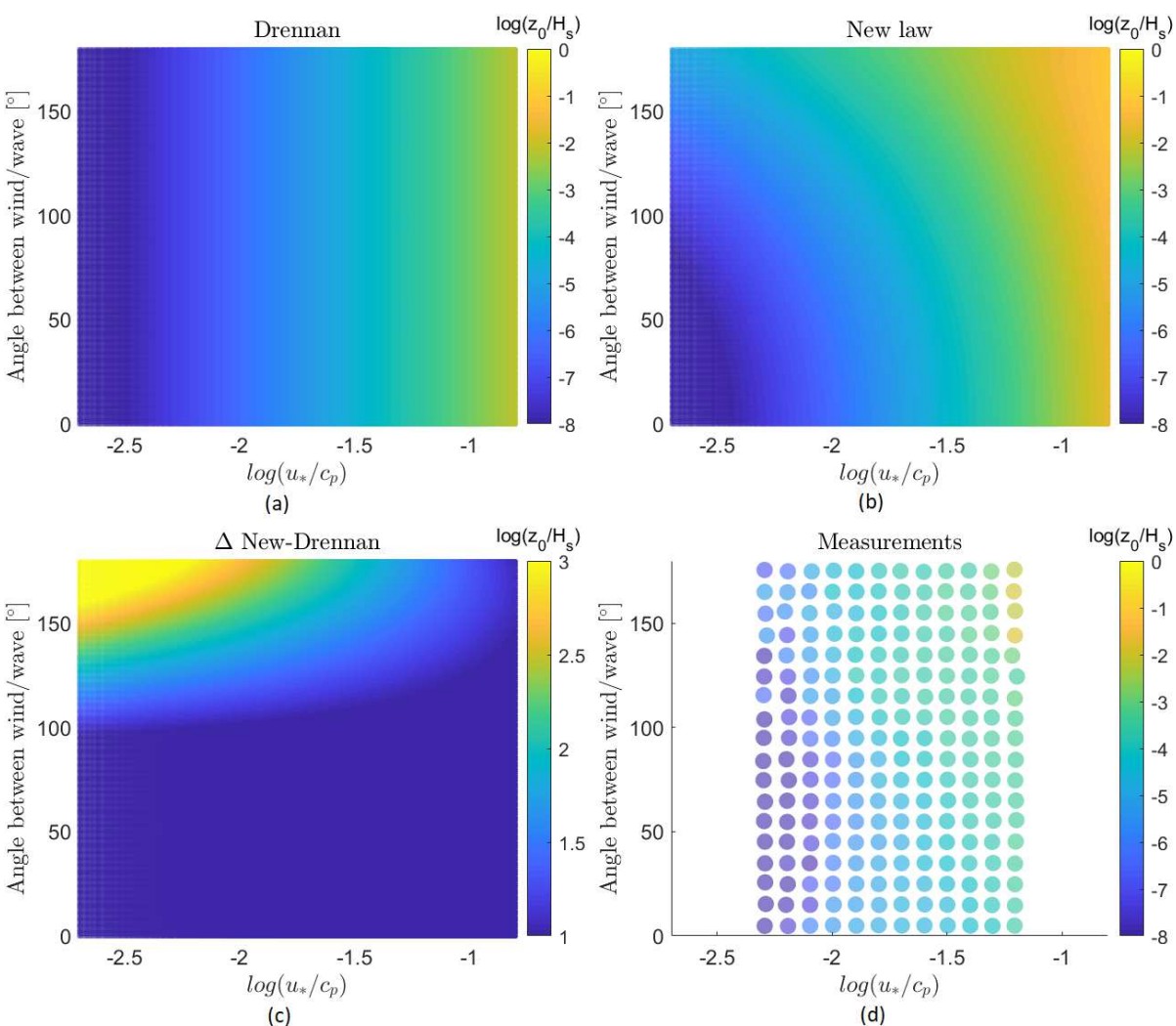

**Figure 9.** (a) Dimensionless roughness length in function of the inverse wave age and the alignment between the wind and wave direction according to Drennan et al. (2003). (b) The new proposed dimensionless roughness length in function of the inverse wave age and the alignment between the wind and wave direction. (c) The differences in roughness length between the law of Drennan et al. (2003) and the new proposed roughness length parameterization in function of the inverse wave age and the alignment between the wind and wave direction. (d) Dimensionless roughness length for the measurements of FINO1 and ASIT as a function of the inverse wave age and alignment between the wind and wave direction.

**Table 4.** Root mean square error for six different alignment classes and for two different roughness length parameterizations: Drennan et al. (2003) and the newly proposed roughness length parameterization.

| $\theta°$ | Drennan et al. (2003) | New roughness length parameterization |
|---|---|---|
| 0° - 30° | 2.35 | 2.34 |
| 30° - 60° | 2.52 | 2.52 |
| 60° - 90° | 2.39 | 2.39 |
| 90° - 120° | 2.58 | 2.46 |
| 120° - 150° | 2.60 | 2.33 |
| 150° - 180° | 2.38 | 2.12 |

tion of Drennan et al. (2003) as a starting point, we investigated the roughness length parameterization of Taylor and Yelland (2001), which takes into account wave steepness. Also for this parameterization, the same effect, namely an increase of the dimensionless roughness length for an increase in misalignment was found. So also this parameterization could be improved by applying a similar methodology as developed here.

A major remark that should be made is whether the use of the Monin-Obukhov similarity theory (MOST) is valid in strong swell cases, as brought forward by Smedman et al. (2009) and Högström et al. (2015). Recently, Li et al. (2018) proposed a modified MOST over water surfaces, based on measurements from a lake. This modified theory includes the relative velocity with respect to the waves, instead of the actual velocity. They suggest that the validity of MOST could improve by using this approach. This new theory, however, is not studied yet for open oceans, as it only has been studied for monochromatic wave

fields occurring on the lake. It is clear that the wind-wave interaction is a complex phenomenon and more research has to be done. This said, most numerical global circulation and mesoscale models still use the variants of the MOST theory with various planetary boundary layer parameterizations. Therefore, keeping MOST as a baseline for our new parameterization, will enable the applicability of our parameterization for various planetary boundary layer parameterizations. Furthermore, upward momentum is not parameterized by the bulk roughness length parameterization proposed in this paper. Notwithstanding that

the inclusion of these points (6% of total data points) did not result in a systematic bias of the newly proposed roughness length parameterization. To include upward momentum, the wave shear stress together with the turbulent shear stress could be imposed instead of a bulk roughness length parameterization. Up to now, however, the bulk parameterization method is used in the majority of numerical mesoscale models, and therefore we base the new parameterization on this.

## 5   Conclusions

In this study we combined two large data sets to investigate the influence of the (mis)alignment between wind and wave direction on the momentum transfer between the sea surface and the atmosphere. We identified a clear difference in roughness length between aligned and opposed wind and wave directions. So far, no roughness length parameterization had been pro-

posed that includes this discrepancy. We used multi-year data from FINO1 and ASIT sites to define a new roughness length parameterization that performs better than state-of-the-art parameterizations.

The new roughness length parameterization can easily be implemented in atmospheric models such as COAWST and WRF (Warner et al., 2010; Skamarock et al., 2008). These mesoscale and microscale atmospheric models can subsequently be used for wind energy assessment studies, estimating the dynamic loads and fatigue as well as the wind turbine wakes, the design of the wind turbine and ultimately the operation of the wind farm (Zeng et al., 1998; Powers and Stoelinga, 2000; Temel et al., 2018). On a larger scale this parameterization is expected to contribute to a better understanding of global wind and wave climates, and global climate change studies in general (Drobynin et al., 2012). While the direct benefits of the new parameterization might seem limited to wind engineering and climate applications, it is also important for a wide range of other applications ranging from the parameterization of optical turbulence over sea, where the vertical fluxes within the surface layer might vary with respect to the wave formations (Frederickcson et al., 2000) to planetary science applications to determine the meteorological conditions for extra-terrestrial atmospheres interacting with lakes, as is the case for Titan (Mitri et al., 2007; Hayes et al., 2013).

*Data availability.* We would like to thank the BMWi (Bundesministerium fuer Wirtschaft und Energie, Fenderal Ministry for Economic Affairs and Energy) and the PTJ (Projekttraeger Jeulich) for providing the FINO1 data, as well as the Woodshole Oceanographic Institution for making the ASIT data available.

## Appendix A:  Wind-wave climate for ASIT and FINO1

An investigation of the dominant wind and wave direction (Fig. A1) at the ASIT measurement mast shows that the main wind direction is southwesterly. Winds coming from the northeastern zone are excluded in the plot, as they lie in the wind shadow zone. The results indicate that wind can come from any direction, while the waves are limited to a sector around south and southwest, which corresponds to the fetch unlimited area south of Martha's Vineyard. No waves are coming from the fetch limited North side.

For the FINO1 measurement mast the main wind direction is southwesterly while the main wave direction is from the fetch unlimited north-west side (Fig. A2). Similar to the ASIT site no waves are traveling from the fetch limited area, in this case south of the measurement mast.

Moreover, aligned wind-wave directions often occurs at higher wind speeds than opposed wind-wave directions (Fig. A3). The waves traveling opposed to the wind direction with a lower wind speed are mostly remotely generated swell moving towards the measurement tower.

For the less frequent occurrences of opposed wind-wave direction the wind is blowing from land while the waves are coming from their fetch unlimited area in the South (Fig. A4), this is for the ASIT tower. The opposite case, where wind is blowing from the sea and waves are coming from land does not occur.

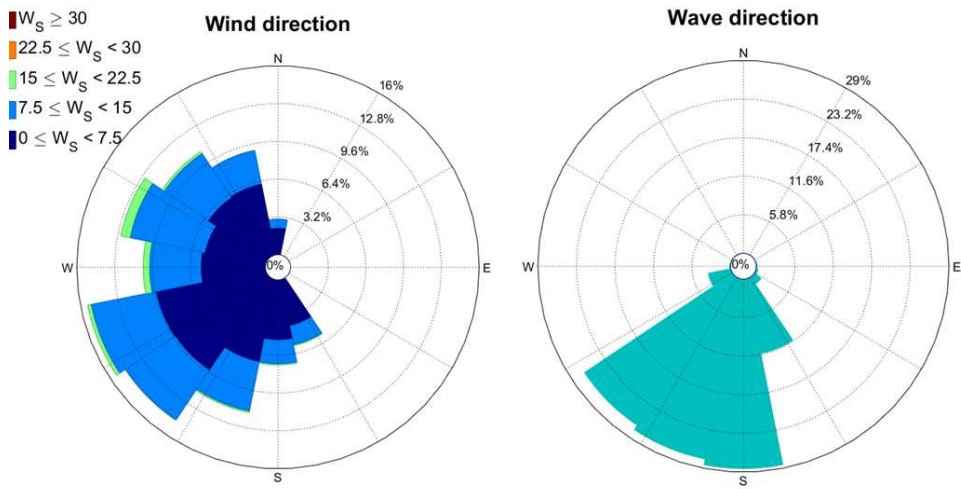

**Figure A1.** On the left side the wind rose excluding the wind shadow zones at z=18 m, while on the right the peak wave direction is shown. Data corresponding to the ASIT measurement mast (65 128 measurement points) during the 2003-2012 period.

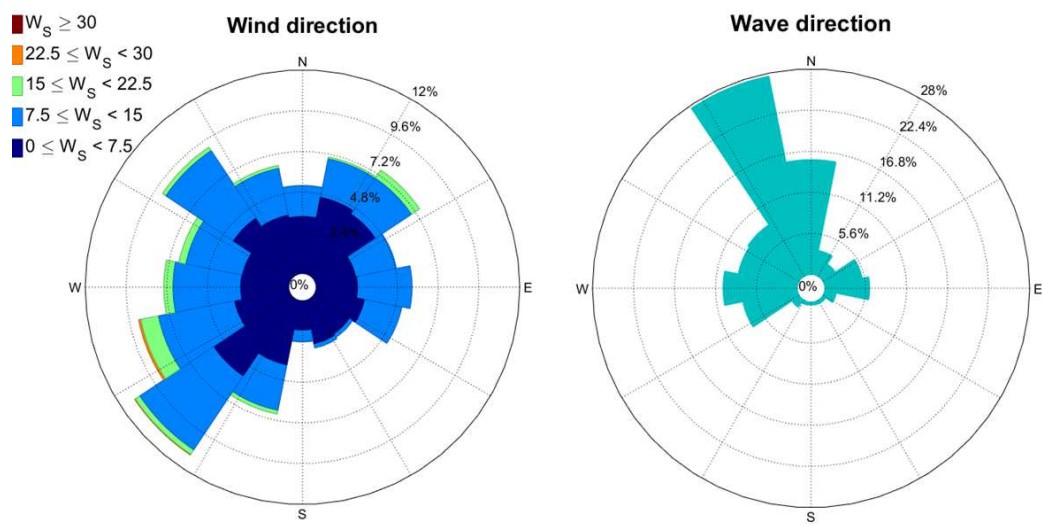

**Figure A2.** On the left side the wind rose excluding the wind shadow zones at z=40 m or z=15 m depending on the measurement campaign, while on the right the peak wave direction is shown. Data corresponding to the FINO1 measurement mast during the year 2010 for z=40m (5 470 measurement points) and 2015-2016 for z=15 m (3 510 measurement points).

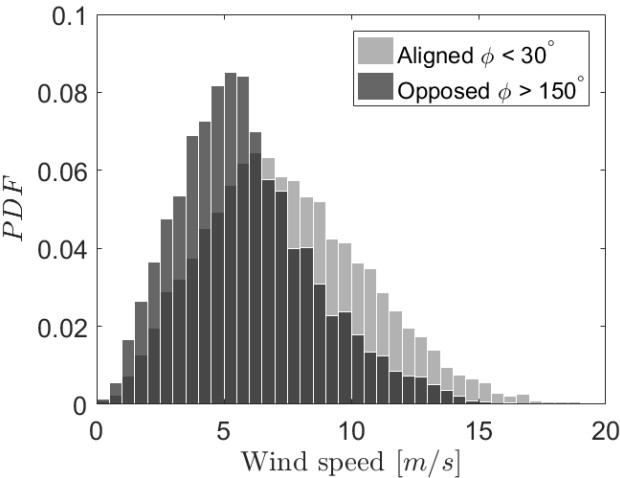

**Figure A3.** Wind speed probability density function (PDF) for aligned wind-wave direction and opposed wind-wave direction, for the ASIT measurement mast.

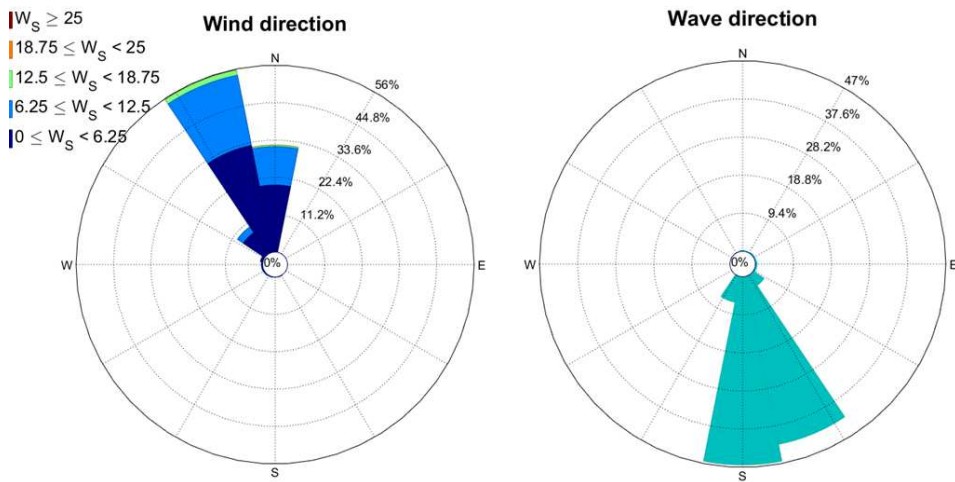

**Figure A4.** On the left side the wind rose excluding the wind shadow zones at z=18m, while on the right the peak wave direction is shown. Data corresponding to the ASIT measurement mast during the 2003-2012 period. This only for wind opposing the wave direction.

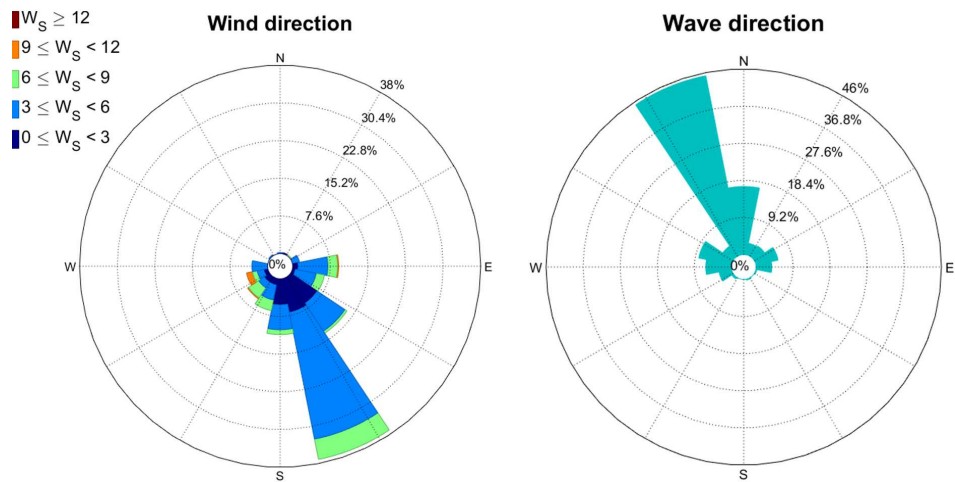

**Figure A5.** On the left side the wind rose excluding the wind shadow zones at z=40 m or z=15 m depending on the measurement campaign, while on the right the peak wave direction is shown. Data corresponding to the FINO1 measurement mast during the year 2010 for z=40m and 2015-2016 for z=15 m. This only for wind opposing the wave direction.

For the FINO1 measurement mast wind-wave misalignment occurs when the waves are traveling from the North-West while the wind is blowing from land in the South-East (Fig. A5).

## Appendix B: Precipitation filter

The FINO1 measurements from the year 2010 include precipitation information at an altitude of 90 m above sea level. The precipitation at 20 m is not used as it was not clear if the measurements were pure precipitation or also sea spray. If precipitation is detected at the location of 90 m above sea level (> 0 mm) then the measurement is excluded from the data set. The comparison between the bin means of the original data set and the data set excluding the precipitation measurements is shown in Fig. B1. No systematic bias is present between data with and without precipitation. For this reason we can conclude that the new roughness length parameterization is not influenced by precipitation.

*Author contributions.* S.P., J.V.B. and N.V.L. conceived and designed the study. Measurements from the OBLEX campaign were obtained by J. R. S.P. and D.M.-E. performed a preprocessing analysis on the FINO1 2010 data. All other analyses were done by S.P. All authors discussed the results and contributed to the manuscript.

*Competing interests.* The authors have no competing interests to declare

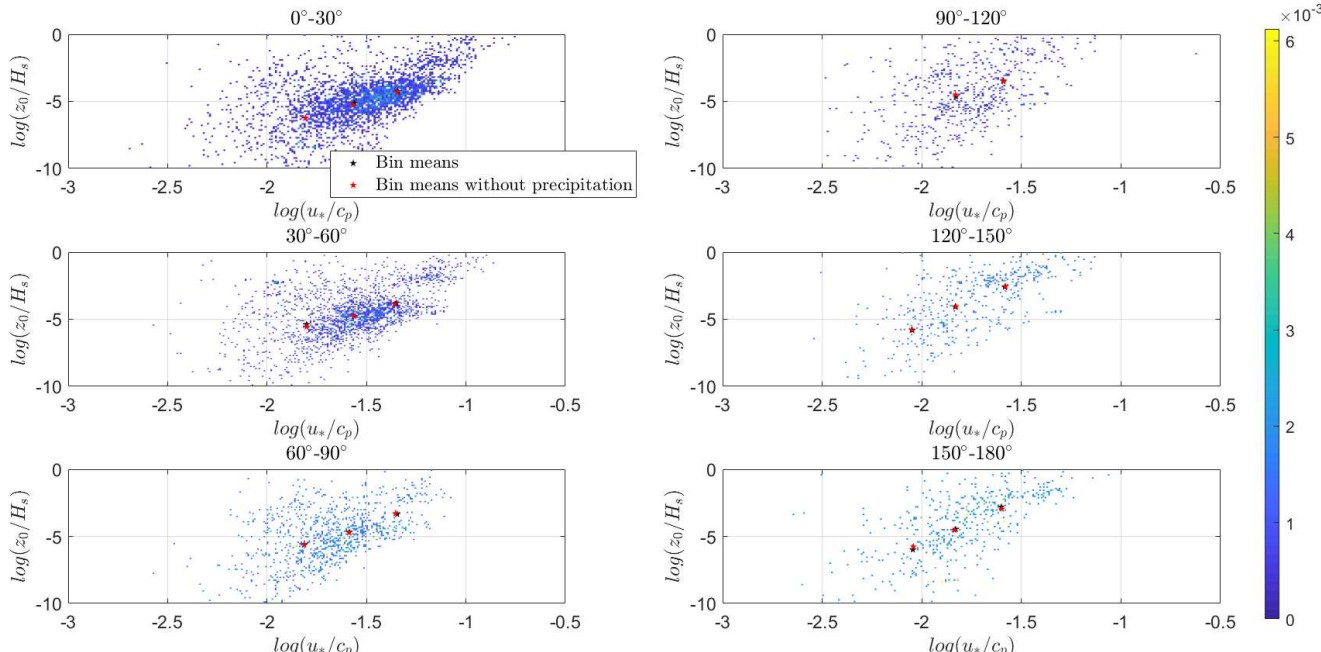

**Figure B1.** The dimensionless roughness length is plotted against the inverse wave age parameter for six different groups of alignment. In each figure the data are bin-averaged, with the bin means (logarithmic) indicated by black stars. While the bin means of the measurement data excluding the precipitation measurements are indicated by red stars. The color scale to the right indicates the probability of occurrence (%) of the measurement points.

*Acknowledgements.* The first author is supported by the Strategic Basic Research grant of the Fonds voor Wetenschappelijk Onderzoek (FWO). We would like to thank the BMWi (Bundesministerium fuer Wirtschaft und Energie, Fenderal Ministry for Economic Affairs and Energy) and the PTJ (Projekttraeger Jeulich) for providing the FINO1 data, as well as the Woodshole Oceanographic Institution for making the ASIT data available. We would like to thank all reviewers for their comments that substantially improved our manuscript.

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
