# Peer review of "A new roughness length parameterization accounting for wind-wave (mis)alignment"

_Atmospheric Chemistry and Physics, 2018_

## Short Comment (SC1) · 20 Sep 2018

Dear authors,

This paper is very interesting, just there are few things which I have not understood. See my comments below.

It is a nice paper overall, very interesting topic. Well done so.

Section1:

- "The wind-wave interaction is located within the Marine Atmospheric Boundary Layer (MABL), which is the lowest part of the atmosphere directly influenced by the sea surface.": could you please clarify/differentiate between the influence of the

seastate (wave spectrum) and the influence of the surface water temperature ? I could imagine that the waves only affect the very lower part of the MABL, see slide 20 of http://www.pcwg.org/proceedings/2014-10-06/06-Turbulence-Intensity-measmnts-offshore-4-PC-verification-wind-res-assmt-R-RiveraLamatA-D-Pollack-Dong.pptx.

- "Contrary to the atmospheric boundary layer (ABL) over land, the effect of the diurnal cycle of the atmospheric stability is negligible due to the high heat capacity of the ocean": could you clarify what you mean by "the effect" ? The influence of stability is still present on the wind profile, though the stability itself varies less (see for instance Figure 4 of http://lr.home.tudelft.nl/fileadmin/Faculteit/LR/Organisatie/Afdelingen_en_Leerstoelen/Afdeling_AEWE/Wind_Energy/Rese _torque_2010.pdf).

- "In addition, higher wind speeds with lower turbulence intensities": lowe than... onshore ?

Section2:

- "The viscous shear stress is assumed to be negligible because of the large scales involved": it is negligible when considering the Reynolds number, right ?

Section3:

- General comment: FINO1 and ASIT are two very different locations with two very different bathymetry and wave climates. Furthermore, the wind and wave datasets are very different too. Could you, in this paper, present the results for both datasets separately ?

- have you considered shoaling and wave height limitation (bottom friction) at the ASIT ? The wave height is depth-limited, and therefore cant grow above a certain threshold at ASIT.

- can you explain more clearly what data have been used, for what directions and in what periods, for what parameters ? maybe in a table... ?

- There are high-quality LiDAR data available at the ASIT, did you know ?
https://www.masscec.com/masscec-metocean-data-initiative

- "The histogram of the angle between the wind and wave direction is shown in Fig. 4". Please make clear that you are using the Peak Wave Direction (see caption of Figure 4). As I understand it, misalignement between WD and Dp does not necessarily mean that the largest waves in the spectrum are swell waves, but instead that swell waves have the most pronounced peak in the spectrum, while the waves can still be wind-driven. In the North Sea especially, there can be 1m swell from North and the rest in Wind-Sea from SW, see https://www.researchgate.net/publication/261834850_Wind_Sea_and_Swell_Waves_in_the_Nordic_Seas.

When you write misalignement, it therefore means most likely "dual seastate", with waves coming from two directions.

- "This behavior is seen for both FINO1 and ASIT measurements.": maybe provide a plot for each ASIT and FINO1 datasets, as the swell conditions are different at both sites. Have you considered wave shoaling at the ASIT ? You can check the offshore waveconditions using https://www.ndbc.noaa.gov/station_page.php?station=44097 and http://cdip.ucsd.edu/?nav=recent&xitem=sfile&stn=154&stream=p1.

- "A first step required is the calculation of the friction velocity" have you tried to correlate u_star with the Turbulence Intensity (std/mean) to check the plausibility of the u_star values ? These should correlate well, see Figure 3 of https://onlinelibrary.wiley.com/doi/full/10.1002/we.1863.

- Using the LiDAR measurement (ASIT) and the mast (FINO1), you could also make a plausibility check of the MOL calculation as in Figure 6 of http://orbit.dtu.dk/files/7872609/2008_01_paper.pdf.

Section4:

- How is cp calculated ? Is it constant for ASIT (shallow water) ? How is L calculated

for FINO1 ?

- Can you just show u* vs z0 and Charnock relationship, to see what the data look like in this dataset ?

- Figure 7: can you make the same plot for 2-3 wind speed ranges and also for 2.3 Hs ranges so we are sure that using non-dimensional analysis works across all speed and scale ranges ?

- General: it is a bit difficult to relate $z\_0/H\_s$ and $u\_star/c\_p$ to what really happens at the site (since the 4 parameters can vary). Maybe some time series plots for selected cases would help understand what happens.
* * *

---

## Referee Comment (RC1) · Anonymous Referee #1 · 17 Oct 2018

This paper analyzed two large datasets of the atmospheric surface layer turbulence measurements and the underlying ocean waves. The authors attempt to come up with a new parameterization for the momentum roughness length zom. The work is very interesting and the datasets are valuable for people interested in air-sea interactions. There are a few points that the authors may want to consider to improve the manuscript:

1. P2, line15, one relevant recent paper you could refer to is : Li, Qi, et al. "Signatures of Air–Wave Interactions Over a Large Lake." Boundary-Layer Meteorology 167.3 (2018): 445-468.
2. P4, " should be function of the wave age and proposed Eq. (10) for the Charnock parameter." should be A function of the wave age ...

[Figure]

3. Throughout the paper, the authors are using roughness to mean roughness length z0m, which I find the single word 'roughness' is a uncommon and it's better to refer to z0m as roughness length.

4. As for the parameterization, would you clarify "This new roughness parameterization, including the alignment of the wind and wave direction, reduces the scatter around the Drennan et al. (2003) parameterization considerably. " on page 16? Which figure shows the reduced scatter? i.e. do you mean plotting log(zom/h) vs. u*/cp and for each different wind-wave angle?

5. For the new parameterization, how to infer if an upward momentum transfer occur? i.e. using the new parameterization in large-scale models and the Monin-Obukhov Similarity Theory, we can compute u*, but we still do not know its direction. Can the authors comment on that?

6. Discussions about the validity of Monin-Obukhov similarity theory under swell condition:such as in Smedman, A., et al. "Observational study of marine atmospheric boundary layer characteristics during swell." Journal of the Atmospheric Sciences 66.9 (2009): 2747-2763. There could be discussions of whether a roughness length is even a valid concept when swell is present. Even though this is not central to your research, it would be good to address such points.

---

## Author Response (AR1)

For clarifying our answers to the reviewers' comments, the following scheme is used: comments of the reviewer are denoted in bold, our answers are denoted in italics and changes in the manuscript are denotes with quotes with the line number and location in the revised manuscript.

We would like to thank all reviewers for their comments that substantially improved our manuscript.

Answers to Mr. Rémi Gandoin

1. **The wind-wave interaction is located within the Marine Atmospheric Boundary Layer (MABL), which is the lowest part of the atmosphere directly influenced by the sea surface.": could you please clarify/differentiate between the influence of the state (wave spectrum) and the influence of the surface water temperature ? I could imagine that the waves only affect the very lower part of the MABL, see slide 20 of http://www.pcwg.org/proceedings/2014-10-06/06-Turbulence-Intensity-meassmnts-offshore-4-PC-verification-wind-res-assmt-R-RiveraLamatA-D-Pollack-Dong.pptx**

*Indeed, both the wave spectrum and the sea surface temperature are characteristics of what we call the sea surface and do influence the MABL. Other physical processes do occur in the MABL and influence it, apart from the wind-wave interaction and sea surface temperature, for example sea spray, breaking waves, streaks … for this reason we did not differentiate between the influence of SST or wind-wave interaction. There are numerical studies that suggest that the wind-wave interaction can be felt up to hub height (Sullivan et al., 2008; Patton et al., 2015; Nilsson et al., 2012). Also Smedman (2009) suggests based on the theory of Miles (1957) that the impact of swell waves extends to a certain height, possibly including the whole atmospheric boundary layer.*

*Rephrased p2 line 8-12* *"The wind-wave interaction mainly occurs within the lowest part of the Marine Atmospheric Boundary Layer (MABL), directly influenced by the sea surface. Numerical studies suggest that the impact of the waves can extent up to hub height, nowadays typically 100 m (Sullivan et al., 2008; Patton et al., 2015; Nilsson et al., 2012). Apart from the wind-wave interaction studied here, there are also other factors affecting the MABL like the sea surface temperature, sea spray, braking waves etc."*

2. **Contrary to the atmospheric boundary layer (ABL) over land, the effect of the diurnal cycle of the atmospheric stability is negligible due to the high heat capacity of the ocean": could you clarify what you mean by "the effect" ? The influence of stability is still present on the wind profile, though the stability itself varies less (see for instance Figure 4 of http://lr.home.tudelft.nl/fileadmin/Faculteit/LR/Organisatie/Afdelingen_en_Leerstoelen/Afdeling_AEWE/Wind_Energy/Research/Publications/Publications_2010/doc/Sathe_-_torque_2010.pdf**

*The "effect" in this sentence implicitly referred to the changes in potential temperature and related velocity changes. We agree that there are still stability variations offshore, however, what we meant is that the typical diurnal cycle present onshore is not present in offshore conditions as a result of the high heat capacity of the ocean. Or for some specific cases (close to the shore) not as strong as onshore (Lapworth, 2005). To avoid confusion, we left out "the effect" from the sentence.*

*Lapworth A. 2005. The diurnal variation of the marine wind in an offshore flow. Q. J. R. Meteorol. Soc.131: 2367–2387*

*Rephrased p2 line 12-13:* "Contrary to the atmospheric boundary layer (ABL) over land, the diurnal cycle of the atmospheric stability offshore is negligible due to the high heat capacity of the ocean."

3.  **In addition, higher wind speeds with lower turbulence intensities": lower than... on-shore?**

*Agree, rephrased p2 line 15-16* "In addition, higher wind speeds with lower turbulence intensities are present for MABLs, which is related to a reduced roughness of the ocean compared to over land."

4.  **The viscous shear stress is assumed to be negligible because of the large scales involved": it is negligible when considering the Reynolds number, right?**

*Indeed, this is what we mean with large scales involved, that we have large Reynolds numbers for the wind flow over the waves.*

*Rephrased p3 line 2:* "The viscous shear stress is assumed to be negligible because of the MABL being characterized by a high Reynolds number".

5.  **General comment: FINO1 and ASIT are two very different locations with two very different bathymetry and wave climates. Furthermore, the wind and wave datasets are very different too. Could you, in this paper, present the results for both datasets separately?**

Thank you for this comment, we added the comment below in the manuscript. See Figure R1, to see the plot for the two locations.

*Added p15 line 5 to p16 line 3:* "It is noted that the bathymetry and wave climates are different for the two measurement locations (FINO1 and ASIT) and that different locations yield different tuning coefficients (Maat et al., 1991; Vickers and Mahrt 1997), and indeed the shape of the scatter plot is slightly different for FINO1 compared to ASIT. On the other hand the two sites show the same pattern with an increasing roughness length for an increasing misalignment. Moreover, the new roughness length parameterization results in a clear improvement for opposed wind and wave directions at both sites. In the end, our parameterization has been proposed targeting implementation in mesoscale models, which can be used to simulate MABL with various wave climates. While it is possible to separate both results, the purpose of this paper was to derive a more general law, taking into account various offshore conditions. Therefore, we decided to group all data and in this way cover a range of conditions as broad as possible. For a better fit at a specific location, the newly proposed parameterization can be tuned according to the data set available. Furthermore, even more different locations and conditions should be included but the availability of simultaneously measured wind and wave parameters is unfortunately scarce."

[Figure]

*Figure R1: The dimensionless roughness length is plotted against the inverse wave age parameter for 3 different groups of alignment for (left) the ASIT measurement mast and (right) for the FINO1 data set. The solid black line represents the roughness parameterization proposed by Drennan et al. (2003), the dashed red line is the new proposed roughness parameterization. The color scale to the right indicates the probability of the occurrence (%) of the measurement points.*

6. **have you considered shoaling and wave height limitation (bottom friction) at the ASIT? The wave height is depth-limited, and therefore can't grow above a certain threshold at ASIT.**

*Wave shoaling has not been taken into account in the analysis of the measurements. In this paper we just want to focus on one aspect of the complex wind-wave interaction. However, we agree that shoaling can influence the roughness length formulation. For this reason, we looked if the effect of an increase in roughness length with an increase of misalignment between the wind and wave direction was present for the roughness length parameterization of Taylor and Yelland (2001), because this parameterization makes use of the wave steepness. See Fig. R2.*

*See also reviewer 2, comment 1.*

*Added p17 line 27-34: "Even though, it is known that multiple parameters play an important role in the complex wind-wave interaction, in this paper we focus on one aspect namely the influence of the difference in direction between the wind and the waves. In order to exclude that the increase of roughness length with increasing misalignment is an artifact of the choice of the roughness parameterization of Drennan (2003) as a starting point, we investigated the roughness length parameterization of Taylor and Yelland (2001), which takes into account wave steepness. Also for this parameterization, the same effect, namely an increase of the dimensionless roughness length for an increase in misalignment was found. So also this parameterization could be improved by applying a similar methodology as developed here.".*

*Added p 17 line 22-23: "Moreover, the depth should be investigated to account for wave shoaling but also to study the effect of bottom friction."*

[Figure]

*Figure R2: The dimensionless roughness is plotted against the wave steepness for 6 different groups of alignment for the ASIT measurement mast and for the FINO1 data sets combined. The solid black line represents the roughness parameterization proposed by Taylor and Yelland (2001). The color scale to the right indicates the probability of the occurrence (%) of the measurement points.*

7. **can you explain more clearly what data have been used, for what directions and in what periods, for what parameters? maybe in a table...?**

Added p9 line 12

| Location | FINO1 | FINO1 | ASIT |
|---|---|---|---|
| Year | 2010 | 2015-2016 | 2003-2012 |
| Wind-shadow | 60°-200° | 245°-360° | 0°-150° |
| High resolution measurement altitude | 40m | 15m | 18m |
| Water depth | 30m | 30m | 15m |

8. **There are high-quality LiDAR data available at the ASIT, did you know ? https://www.masscec.com/masscec-metocean-data-initiative**

*Thank you, we were not aware of these measurements. We intend to use them in future studies.*

9. **The histogram of the angle between the wind and wave direction is shown in Fig. 4". Please make clear that you are using the Peak Wave Direction (see caption of Figure 4). As I understand it, misalignement between WD and Dp does not necessarily mean that the largest waves in the spectrum are swell waves, but instead that swell waves have the most pronounced peak in the spectrum, while the waves can still be wind-driven. In the North Sea especially, there can be 1m swell from North and the rest in Wind-Sea from SW, see https://www.researchgate.net/publication/261834850_Wind_Sea_and_Swell_Waves_in_the_Nordic_Seas.**

*Indeed, we did check the WW3 partitioned data of the FINO measurement mast and could see that we sometimes had 2 wave systems, swell waves from the North and wind-sea from the South-West. However, most of the time in these cases the peak wave direction corresponded to the swell wave direction.*

*Agree, rephrased p9 line 21-22: "The histogram of the angle between the wind and peak wave direction is shown in Fig 4." Also in caption of Fig 4 (rephrased p9 line 1).*

**10. This behavior is seen for both FINO1 and ASIT measurements.":  maybe provide a plot for each ASIT and FINO1 datasets, as the swell conditions are different at both sites.  Have you considered wave shoaling at the ASIT? You can check the offshore wave conditions using https://www.ndbc.noaa.gov/station_page.php?station=44097 and http://cdip.ucsd.edu/?nav=recent&xitem=sfile&stn=154&stream=p1**

*Indeed, we do have histograms for both sites (Fig. R3). For both the ASIT and the FINO1 measurement masts there can be seen that wind-wave direction alignment ($\theta < 30°$) occurs most of the time, while opposed wind wave directions ($\theta > 150°$) is less frequent. (Considering wave shoaling see question 6).*

*Added p9 line 25 to p10 line 3: "This behavior is seen for both FINO1 and ASIT measurements. The difference between ASIT and FINO1 is that for the ASIT measurement mast 90° misalignments are more present than for the FINO1 measurements, this could be due to more fetch unlimited areas at ASIT."*

[Figure]

*Figure R3: Probability density function (PDF) of the angle between the wind and wave direction for (up)the ASIT measurement mast and (down) for the FINO1 mast.*

**11. A first step required is the calculation of the friction velocity" have you tried to correlate u_star with the Turbulence Intensity (std/mean) to check the plausibility of the u_star values ? These should correlate well,  see Figure 3 of https://onlinelibrary.wiley.com/doi/full/10.1002/we.1863**

*You are right that the friction velocity should correlate with the turbulence intensity. However, the friction velocity in our study is calculated based on the eddy covariance technique on the high resolution measurements (following Drennan et al., 2005) and not based on the log law. For this reason, we believe that checking the validity of the friction velocity would mean the same as checking the validity of the turbulence intensity and would not give us more information. We think that an analysis of the turbulence*

*intensity could help us in further studies to understand the turbulent kinetic energy budget at the surface and improve surface layer schemes.*

**12. Using the LiDAR measurement (ASIT) and the mast (FINO1), you could also make a plausibility check of the MOL calculation as in Figure 6 of http://orbit.dtu.dk/files/7872609/2008_01_paper.pdf**

*Thank you for mentioning the LIDAR measurements which we were not aware of. Investigating this is out of the scope of this paper. Based on various peer-reviewed publications such as Patton et al. (2015), Barthelmie (1999), Edson et al. (2007) and Crofoot (2004) we proceeded without performing any additional plausibility check as suggested. The final dimensional plots of z0/Hs vs u_star/c_p do also correspond to what is known from literature (Drennan et al., 2003).*

**13. How is cp calculated? Is it constant for ASIT (shallow water)? How is L calculated for FINO1?**

*The cp calculation before was just calculated based on shallow water or deep water limits of the wave phase speed. We modified the formulation to the full dispersion relation for the wave phase speed calculation, as suggested by the reviewer, to be more correct. So the transitional depths are also taken into account. For this reason, the figures 5/8 and 9 changed in the manuscript as did the coefficients of the new roughness length parameterization:*

$$\frac{z_0}{H_s} = 20 \cos(0.45\,\theta) \left(\frac{u_*}{c_p}\right)^{3.8\cos(-0.32\,\theta)}$$

*We could see that in most cases the ASIT measurement location would correspond to shallow water, while in the FINO1 measurement location the deep water condition did occur more often. For the FINO1 location the peak wave period is measured and used to calculate the wave speed.*

*L for the FINO1 data was already calculated by Munoz-Esparza et al. (2012) for the 2010 data set. For the 2015-2016 data set it was calculated by eq (16) were the virtual potential temperature and the temperature flux were obtained by the EddyPro software. (p 11 line 12-14).*

*Added p4 line 16-17 "For the calculation of the wave phase speed the full dispersion relation has been used in this study."*

*Changed Fig 5 (p 13), Fig 8 (p 16), Fig 9 (p 18).*

*Changed formula (20) (p 15)*

**14. Can you just show u* vs z0 and Charnock relationship, to see what the data look like in this dataset?**

*In Fig. R4 (a) you can see the Charnock relation together with the measurement points of both the ASIT and the FINO1 measurement locations. In Fig R4 (b) you can see the root mean square error of the measurement points compared to the Charnock parameterization. This Charnock parameterization is used in many numerical mesoscale codes (WRF, COSMO, HARMONIE ….). However, it has been recognized that for offshore conditions a constant Charnock parameter is not sufficient. The sea state should be included, and different approaches have been developed, Drennan (2003), Taylor Yamada (2001), …. You can see that the spread of the Charnock relation is larger than this for Drennan (2003) (see Fig 5(a) in the*

*manuscript). Furthermore, in this manuscript we decreased the spread around the roughness law even more by taking the impact of the direction into account. This is just a starting point to improve the roughness length parameterization even more for the numerical models.*

*Added p 12 line 8-14**: "The same validation can be done for the Charnock roughness length parameterization, used in many numerical mesoscale codes, by plotting the roughness length against the friction velocity. However, multiple studies like Drennan et al. (2003), Taylor and Yelland (2001), Janssen (1991), Fan et al. (2012) and Liu et al. (2011) among many others recognized to include more information about the sea state as this improves the estimation of the roughness length. As these studies were a starting point to improve the Charnock's roughness length parameterization for numerical models by including extra information from the sea state, our goal is to build further upon this by taking into account the alignment between the wind and wave direction."*

[Figure]

*Figure R4: (a) The roughness is plotted against the friction velocity. The black line is the Charnock parameterization with $\alpha = 0.018$. The color scale to the right indicates the probability of occurrence (%) of the measurement point of both ASIT and FINO1 data sets. (b) The root mean square error of the measurement points compared to the Charnock parameterization.*

15. **Figure 7: can you make the same plot for 2-3 wind speed ranges and also for 2.3 Hs ranges so we are sure that using non-dimensional analysis works across all speed and scale ranges?**

*See next question*

16. **General: it is a bit difficult to relate z_0/H_s and u_star/c_p to what really happens at the site (since the 4 parameters can vary). Maybe some time series plots for selected cases would help understand what happens.**

*In order to be consistent with the existing literature on the offshore roughness parameterizations, we have used the referred variables same as was first introduced by Hsu (1973) and used by many other authors like Donelan (1990), Drennan (2003), Patton et al. (2015) and is used in different numerical models by Warner et al. (2010) and Bolanos et al. (2014).*

We would like to thank the reviewer for his/her comments that substantially improved our manuscript.

Answers to Referee 1

1. **P2, line15, one relevant recent paper you could refer to is : Li, Qi, et al. "Signatures of Air–Wave Interactions Over a Large Lake." Boundary-Layer Meteorology 167.3 (2018): 445-468**

*Added reference p2 line 18*

*Added p19 line 2-5* *"Recently, Li et al. (2018) proposed a modified Monin-Obukhov similarity theory over water surfaces, based on measurements from a lake. This modified theory includes the relative velocity with respect to the waves, instead of the actual velocity. They suggest that the validity of the Monin-Obukhov similarity theory could improve by using this approach. This new theory, however, is in not studied yet for open oceans, as it only has been studied for monochromatic wave fields occurring on the lake."*

2. **P4, " should be function of the wave age and proposed Eq. (10) for the Charnock parameter." should be A function of the wave age ...**

Corrected p5 line 5-6 *"should be a function of the wave age and proposed Eq. (10) for the Charnock parameter."*

3. **Throughout the paper, the authors are using roughness to mean roughness length z0m, which I find the single word 'roughness' is a uncommon and it's better to refer to z0m as roughness length**

*Corrected within the whole manuscript. (roughness → roughness length)*

4. **As for the parameterization, would you clarify "This new roughness parameterization, including the alignment of the wind and wave direction, reduces the scatter around the Drennan et al. (2003) parameterization considerably. " on page 16? Which figure shows the reduced scatter? i.e. do you mean plotting log(zom/h) vs. u\*/cp and for each different wind-wave angle?**

*The Root Mean Square Error is calculated for the different groups of alignment. This is done for the parameterization of Drennan et al. (2003) and for the newly proposed roughness length parameterization compared to the measured values. The results are shown in Table 1 where a decrease in RMSE is present for the newly proposed roughness length parameterization compared to the law of Drennan et al. (2003) for misaligned wind wave direction, while they perform equally well for aligned wind-wave directions.*

*Added p19 line 1; A summary of these results is presented in Table 1 and added in the manuscript.*

*Table 1: RMSE for six different alignment classes and for two different roughness length parameterizations: Drennan et al. (2003) and the newly propose roughness length parameterization.*

| Alignment class | Drennan et al. (2003) | New roughness parameterization |
|---|---|---|
| 0°-30° | 2.35 | 2.34 |
| 30°-60° | 2.52 | 2.52 |
| 60°-90° | 2.39 | 2.39 |
| 90°-120° | 2.58 | 2.46 |
| 120°-150° | 2.60 | 2.33 |
| 150°-180° | 2.38 | 2.12 |

**5. For the new parameterization, how to infer if an upward momentum transfer occur? i.e. using the new parameterization in large-scale models and the Monin-Obukhov Similarity Theory, we can compute u\*, but we still do not know its direction. Can the authors comment on that?**

*For our data set upward momentum transfer occurred 6% of the time. First we investigated if the use or exclusion of these points would change something regarding the newly proposed roughness length parameterization. In Fig. R5 (left) you can see the results including the upward momentum and (right) excluding these points. So there is no systematic error by including or excluding these points.*

*Grachev and Fairall (2001) concluded that "upward momentum reaches zero around a wind speed U ~ 1.5–2 m/s, which corresponds to wave age cp /U ~ 10. And a further decrease of wind speed results with a sign reversal of momentum flux". If only an atmospheric model like WRF is used it is only possible to estimate whether a sign reversal occurs in the momentum flux. However, in the case of a coupled atmosphere wave model, the shear stress due to the wave could be calculated in addition to the turbulent shear stress and as such the momentum transfer would be calculated.*

*Grachev, A. A., and C. W. Fairall. "Upward momentum transfer in the marine boundary layer." Journal of physical oceanography 31.7 (2001): 1698-1711.*

*Added p19 line 8-13**: "Furthermore, upward momentum is not parameterized by the bulk roughness length parameterization proposed in this paper. Notwithstanding that the inclusion of these points (6% of total data points) did not result in a systematic bias of the newly proposed roughness length parameterization. To include upward momentum, the wave shear stress together with the turbulent shear stress could be imposed instead of a bulk roughness length parameterization. Up to now, however, the bulk parameterization method is used in the majority of numerical mesoscale models, and therefore we base the new parameterization on this."*

[Figure]

*Figure R5: The dimensionless roughness is plotted against the inverse wave age parameter for 3 different groups of alignment for (left) measurements including upward momentum and (right) measurements excluding upward momentum. The solid black line*

*represents the roughness parameterization proposed by Drennan et al. (2003), the dashed red line is the new proposed roughness parameterization. The color scale to the right indicates the probability of the occurrence (%) of the measurement points.*

6. **Discussions points. about the validity of Monin-Obukhov similarity theory under swell condition:such as in Smedman, A., et al. "Observational study of marine atmospheric boundary layer characteristics during swell." Journal of the Atmospheric Sciences 66.9 (2009): 2747-2763. There could be discussions of whether a roughness length is even a valid concept when swell is present. Even though this is not central to your research, it would be good to address such points.**

*Indeed, this is not stressed enough in our manuscript and we changed accordingly.*

*Added p19 line 1-8: "A major remark that should be made is whether the use of the Monin-Obukhov similarity theory (MOST) is valid in strong swell cases, as brought forward by Smedman et al. (2009) and Högström (2015). Recently, Li et al. (2018) proposed a modified MOST over water surfaces, based on measurements from a lake. This modified theory includes the relative velocity with respect to the waves, instead of the actual velocity. They suggest that the validity of MOST could improve by using this approach. This new theory, however, is not studied yet for open oceans, as it only has been studied for monochromatic wave fields occurring on the lake. It is clear that the wind-wave interaction is a complex phenomenon and more research has to be done. This said, most numerical global circulation and mesoscale models still use the variants of the MOST theory with various planetary boundary layer parameterizations. Therefore, keeping MOST as a baseline for our new parameterization, will enable the applicability of our parameterization for various planetary boundary layer parameterizations."*

We would like to thank the reviewer for his/her comments that substantially improved our manuscript.

Answer to referee 2

**Since there are not so many research papers dedicated to the directional aspects of the sea surface roughness, the subject of the manuscript was interesting and promising. Unfortunately the handling of the subject was rather superficial.**

**The manuscript is a straightforward attempt to parameterise all kind of cases into a mean behaviour taking into account the wave age and difference in dominant wave and wind directions. The parameterisation might work as a practical solution to include all situations but I am not convinced that it will improve e.g. the estimates of the vertical wind profiles, given the observed different wind profile during swell.**

*We present the new parameterization as a starting point for a better representation of directional aspects of the wind-wave interactions in numerical models. We are well aware that it is not a full representation of all situations, however the parameterizations used up to now do contain even less information, i.e. they do not include the difference in direction between the wind and the peak wave direction, which is known from literature (Grachev et al., 2003; Sullivan et al., 2000; Kalvig et al., 2013 …) to be a major limitation of existing parameterizations.*

*The comments of the reviewer also made us realize that the goal of our work was perhaps not explicit enough and we improved that by adding the following text:*

*Added p2 line 21-24 : "The goal of this paper is threefold, first we would like to identify from observations if the roughness length is dependent on the alignment between the wind and peak wave direction. Second, we aim at developing a method to include this alignment effect in atmospheric models. Third, we apply the method mentioned before to derive a specific parameterization for the atmospheric models based on the limited existing observations. The parameterization can subsequently be improved when more data become available."*

*We are currently working on the implementation of the newly proposed roughness length parameterization in the COAWST model. Although we feel that the current manuscript should stay focused on the presentation of the parameterization, we would like to take the opportunity to show our preliminary result to the reviewer. We therefore added an appendix to this reply discussing this preliminary analysis.*

**The benefit of the study is to stress the importance of the directional information, but that alone is not really a new result as it has been brought up in the literature cited in the manuscript. In this manuscript, though, the amount of data is considerable. Unfortunately the study in its present form does not contribute much to our understanding of the complex interplay between the waves and atmosphere.**

*In retrospect the goal of our work was perhaps not explicit enough. See previous comment.*

*Modified p1 line 9-11: "Using this new roughness parameterization in numerical models might facilitate a better representation of offshore wind, which is relevant to many applications including offshore wind energy and climate modeling."*

*Modified p7 line 1-3: "With these results we propose a new roughness length parameterization, to support future research on the improvement of numerical modeling of the transfer of momentum between the sea surface and the atmosphere, in particular for opposed wind-wave directions."*

*Modified p19 line 20 to page 20 line 4: "The new roughness length parameterization can easily be implemented in atmospheric models such as COAWST and WRF (Warner et al., 2010; Skamarock et al., 2008). These mesoscale and microscale atmospheric models can subsequently be used for wind energy assessment studies, estimating the dynamic loads and fatigue as well as the wind turbine wakes, the design of the wind turbine and ultimately the operation of the wind farm (Zeng et al., 1998; Powers and Stoelinga, 2000; Temel et al., 2018). On a larger scale this parameterization is expected to contribute to a better understanding of global wind and wave climates, and global climate change studies in general (Drobynin et al., 2012)."*

**It is regrettable, since the authors have good material to do a more detailed analysis of the momentum flux and the impact of the directional aspects.**

1. **There was not much analysis of the two datasets used in the study: the only criteria was that there was not shadowing of the mast structures on the atmospheric side.**
   **No consideration was given:**
   - **the impact of the water depth to the wave field: changes in wave steepness, depth induced breaking, refraction etc, all contributing to the sea surface roughness. Especially the ASIT wave data is from shallow water, but the 30 m depth at FINO1 is not free from these additional factors either**
   - **to other reasons for the differences in wind and dominant wave directions than swell due to possible refraction or slanting fetch cases, especially at the location of ASIT**

*We are aware that there are multiple effects potentially playing a role at the interface between the atmosphere and the sea surface influencing the roughness length. Although additional analysis on these effects would be interesting we do not have the data to do so.*

*Having said this, we can do an analysis of the importance of different effects. Firstly, to analyze depth induced breaking we did a back of the envelope calculation which tells us that depth induced breaking will not occur at both sites for the available time period. For swell waves at the ASIT location combined refraction and shoaling would not lead to larger wave heights for wave periods less than 15 seconds, and wave breaking would therefore only start at swell height larger than 5-6m, values which do not occur in the measurements. Secondly, a change of direction of more than 20° would occur for a wave of 15 seconds if the angle between the normal to the coastline and the wave direction is more than 45°. For all shorter periods this would be less. For the ASIT measurements this would mean that less than 20% of the swell records could be affected. Although limited in occurrence, this effect might contribute to the scatter observed.*

*Wind waves with periods lower than 5.5 seconds (kh=2) can be considered as nearly deep water waves constitute 45% of the records and therefore the possible impact of bottom friction will be minimal. For the other wind wave records some effect of depth limitation might occur, but depth induced breaking would only occur for significant wave heights larger than about 6m, which is not the case for the measurements that were used.*

*Although some of these different effects have been studied, most papers focus on one element without combining all factors. Different authors focus on different aspect of this complex interaction. For example, Jiminez and Dudhia (2018) only look at the influence of the depth and propose a modified roughness length parameterization while acknowledging other aspects, such as fetch or wave age to have an effect. Similarly, Liu et al. (2011) use a parameterization taking into account wave age and sea-spray while not taking into account for example the effect of wave direction and depth. In order to exclude that the increase of roughness length with increasing misalignment is an artifact of the choice of the roughness parameterization of Drennan (2003) as a starting point, we investigated the roughness length parameterization of Taylor and Yelland (2001), which takes into account wave steepness. Also for this parameterization, the same effect, namely an increase of the dimensionless roughness length for an increase in misalignment was found (see Fig. R6).*

*Added p17 line 28-34: "Even though, it is known that multiple parameters play an important role in the complex wind-wave interaction, in this paper we focus on one aspect namely the influence of the difference in direction between the wind and the waves. In order to exclude that the increase of roughness length with increasing misalignment is an artifact of the choice of the roughness parameterization of Drennan (2003) as a starting point, we investigated the roughness length parameterization of Taylor and Yelland (2001), which takes into account wave steepness. Also for this parameterization, the same effect, namely an increase of the dimensionless roughness length for an increase in misalignment was found. So also this parameterization could be improved by applying a similar methodology as developed here.".*

[Figure]

*Figure R6: The dimensionless roughness is plotted against the wave steepness for 6 different groups of alignment for the ASIT measurement mast and for the FINO1 data sets combined. The solid black line represents the roughness parameterization proposed by Taylor and Yelland (2001). The color scale to the right indicates the probability of the occurrence (%) of the measurement points.*

*Having said this, we now also realize that our manuscript was not explicit enough detailed with the limitations of our work. We thank the reviewer for pointing this out and modified the manuscript accordingly:*

*Modified p17 line 14-26: "The remaining scatter indicates, however, that not all relevant physical processes occurring in the MABL are adequately described by the roughness length parameterization, leaving room for future improvements. Liu et al. (2011) found that sea spray, again an interplay between wind and wave, also influences the roughness length. Furthermore, not only sea spray but also wave steepness of swell waves alter the momentum transfer between the sea and the atmosphere, which in turn influences the roughness length. The wind stress decreases if the swell steepness increases (Ocampo-Torres et al., 2011). In fact, García-Nava et al. (2012) proposed a new roughness length parameterization which includes both the effect of the wave age and the swell steepness on the roughness length. Recently, Jiménez and Dudhia (2018) also found that the roughness length parameterization should be adapted considering the depth present. Moreover, the depth should be investigated to account for wave shoaling but also to study the effect of bottom friction. As such, future work of the combined effect of wind-wave misalignment and the effect of sea spray, swell steepness and depth, are needed to further improve the roughness length parameterization for numerical models. This requires additional observational data to be taken, for example, the swell height and sea spray information were not available for our measurement sites at this moment in time."*

- **and most importantly, to the structure of the MABL during swell. The difference between the wind and dominant wave direction was mainly attributed to the swell cases/mixed seas with a dominant swell. There are several papers about the changes in the MABL when a swell is present, see for example the series of papers by Smedman et al. and Högström et al. and the references within. The validity of Monin-Obukhov scaling, and the existence of the logarithmic profile is questionable when there is swell present. There is no discussion about the different characteristics caused by a swell in the manuscript, or if the presented analysis is suitable of handling the subject in the first place**

*Indeed, this is not stressed enough in our manuscript and we changed accordingly. Also reviewer one raised this point while at the same time acknowledging that this is not the central point of the research.*

*Added p19 line 1-8: "A major remark that should be made is whether the use of the Monin-Obukhov similarity theory is valid in strong swell cases, as brought forward by Smedman et al. (2009) and Högström (2015). Recently, Li et al. (2018) proposed a modified Monin-Obukhov similarity theory over water surfaces, based on measurements from a lake. This modified theory includes the relative velocity with respect to the waves, instead of the actual velocity. They suggest that the validity of the Monin-Obukhov similarity theory could improve by using this approach. This new theory, however, is not studied yet for open oceans, as it only has been studied for monochromatic wave fields occurring on the lake. It is clear that the wind-wave interaction is a complex phenomenon and more research has to be done. This said, most numerical global circulation and mesoscale models still use the variants of the MOST theory with various planetary boundary layer parameterizations. Therefore, keeping MOST as a baseline for our new parameterization, will enable the applicability of our parameterization for various planetary boundary layer parameterizations."*

2. **The Drennan et al. 2003 parameterisation is for pure wind-sea cases: the swell and mixed sea cases where carefully excluded. There is not really reason to expect that the parameterisation would be valid in all kind of situations. This is clear also from the comparison paper by Drennan et al. 2005. The latter paper even recommended to use Smith (1980) parameterisation for swell cases. The results were not compared to the model calculations of Patton et al. 2015 which would have been closer to the way of handling the subject in this manuscript**

*Indeed, Drennans (2003) roughness length parameterization is developed for pure-wind sea cases. However, it is used in most numerical models like WRF, COAWST, COSMO, WRF-MIKE21, … for different offshore conditions. Most numerical models even use the Charnock parameterization which does take the sea state parameters as a constant. Therefore, we took Drennan's (2003) parameterization as a starting point to improve the parameterization by including the dependency on the directional difference between the wave and the wind. This method has been chosen because Drennan's parameterization takes into account fetch and duration which is an important factor in most areas offshore and because it is one of the most used parameterization in coupled atmosphere wave models.*

*Indeed, comparing the results with Patton could have been in option. However, it is not finalized, the constant A is not known. Moreover, looking at their LES results the roughness length parameterization proposed by them follows a similar trend as ours where there is an increase in the roughness length for an increase in misalignment between the wind and wave direction.*

*Mentioned p 6 line 21-27**: Unfortunately, this new roughness length parameterization is not finalized yet. For young wave ages, unrealistic dimensionless roughness length values are obtained and the constant A is undefined. Additionally, this parameterization has only been tested on the results of LES simulations. Lastly, these simulations included only imposed waves and a one-way wind-wave interaction. As such, only the effects of the waves on the wind, were studied. Important here is that Drennan et al. (2003) and Patton et al. (2015) suggested that a new roughness length parameterization should include the angle between the wind and the wave direction, consistent with the wind profiles obtained by the LES and RANS simulations of Sullivan et al. (2000) and Kalvig et al. (2013).*

*In our paper we decide not to use the Smith (1980) model as a starting point, because it is not often used in coupled atmosphere-wave models. Moreover, we decided to use the formula of Drennan (2003) as a starting point as for most cases it is better than the formula of Smith (1980).*

3. **p.4, Equations 5 and 6: why the full equation for phase speed was not used to cover the intermediate depths?**

Modified

*The cp calculation before was just calculated based on shallow water or deep water. However, we modified the formulation to the full dispersion relation for the wave phase speed calculation. So the transitional depths are also taken into account. For this reason, the figures 5/8 and 9 changed in the manuscript as did the new roughness length parameterization:*

$$\frac{z_0}{H_s} = 20 \cos(0.45\,\theta)\left(\frac{u_*}{c_p}\right)^{3.8\cos(-0.32\,\theta)}$$

*We could see that in most cases the ASIT measurement location would correspond to shallow water, while in the FINO1 measurement location the deep water condition did occur more often. For the FINO1 location the peak wave period is measured and used to calculate the wave speed.*

*Added p4 line 15-16 "For the calculation of the wave phase speed the full dispersion relation has been used in this study."*

*Changed Fig 5 (p 13), Fig 8 (p 16), Fig 9 (p 18).*

*Changed formula (20) (p 15)*

**4. P. 5, Lines 24-25: Drennan et al. 2005 discussed how the swell direction affected the sea surface roughness, but they did not present a parameterisation that accounts for the swell and its direction**

*Agree, what we wanted to say is that he suggested to have a parameterization including the direction between the wind and wave direction.*

*Rephrased p6 line 6-8: "However, these roughness length parameterization performed poorly in regions of swell and Drennan (2003) suggested that a more elaborated roughness length parameterization including not only the swell magnitude but also the direction of the swell waves could improve the model."*

*Rephrased p6 line 13-14: "While a wind-wave direction based roughness length parameterization has not yet been investigated, the importance of this effect has been suggested by Drennan et al. (2005) based on experimental observations."*

**5. P. 6, Section 3. Methods. There is no mention who runs FINO1 and ASIT.**

*This is mentioned in the data availability.*

*P 20 line 9-11: We would like to thank the BMWi (Bundesministerium fuer Wirtschaft und Energie, Fenderal Ministry for Economic Affairs and Energy) and the PTJ (Projekttraeger Jeulich) for providing the FINO1 data, as well as the Woodshole Oceanographic Institution for making the ASIT data available.*

**6. P.7. Lines 11-12. The location of the wave buoy is not mentioned.**

*Rephrased p 8 line 9 -10: "Wave information, including significant wave height, peak wave direction and (peak) wave period are measured by a Datawell/MKIII buoy in the close vicinity of the FINO1 measurement mast."*

**7. P.9, Line 13. What does wind-wave equilibrium mean? Same directions? There are some other loosely used terms in the manuscript as well, e.g. wave direction -> mean wave direction at the spectral peak.**

*Wind-wave equilibrium is indeed not necessarily the case if the wind and wave direction are aligned.*

*Rephrased p 9 line 23-25: "Wind-wave direction alignment ($\theta < 30\circ$) occurs most of the time (33%), however, as can be seen from Fig. 4, there is significant probability of occurrence of misalignment events of different degrees, while opposed wind and wave directions ($\theta > 150\circ$) is a less frequent scenario (8%).*

*Roughness → changed to roughness length*

*Rephrased p3 line 7-8: ".. is the wave phase speed at the peak of the wave energy spectrum, hereafter referred to as wave speed".*

*Wave direction → changed to peak wave direction*

*Rephrased p 6 line 12-13: "The focus of this paper, however, will be on the influence of the difference between the peak wave direction, hereafter referred to as wave direction, and the wind direction on the roughness length parameterization."*

8. **P. 15 Fig. 8. The scatter is still large. There probably are other reasons as well, not just the wave age and difference in wind and wave directions**

*Agree, see point 1*

Appendix

*Some preliminary simulations have been made, namely a ten-day simulation (09-12-2010 to 19-12-2010) over the FINO1 location. The domain of the simulation can be seen in Fig. R7, which shows 3 nested domains for WRF (atmospheric model) and 2 nested domains for SWAN (wave model). The resolution on the three WRF domains are 27 km, 9 km and 3 km and 100, 93, 93 grid points for the largest, medium and smallest domain respectively. In the vertical direction 53 levels were used. For the boundary conditions and initial conditions, we use ERA-INTERIM data with a resolution of 0.72° and a 6 hourly update. For the wave model the resolution of the grids is 20 km and 3 km with 100 and 173 grid points in each direction for the largest and smallest domain respectively. The results of the simulations are compared to the FINO1 measurements.*

*The results of these preliminary simulations are shown in Fig. R8 (a). It was found that the mean velocity profile of the newly proposed roughness length parameterization approximates the measurements better than the results of the simulation with the roughness length parameterization of Drennan (see also Fig. R8 (b)), but differences are small and statistical tests indicate that the results from the new parameterization are not significantly different from the one of Drennan (p-value 0.23), while the newly proposed roughness length parameterization and Drennans (2003) roughness length parameterization are significantly different from the results of WRF only (p-value 0.002). For this reason, we are running a longer simulation of one year. The strongest decrease is found for misaligned wind and wave direction if we look to the results based on different degree of alignment (Fig. R9).*

[Figure]

*Figure R7: Domain set up for COAWST simulations centered around the FINO1 location. The black lines represent the atmospheric domains, which are three nested domains. The red lines represent the nested wave domains.*

[Figure]

*Figure R8: (a) Mean velocity profile of (1) the FINO1 measurements together with one standard deviation (2) the results of an atmospheric (WRF) only simulation (3) the results of a coupled atmospheric (WRF) wave (SWAN) model using Drennans roughness parameterization and (4) the results of a coupled atmospheric (WRF) wave (SWAN) model using the newly proposed roughness parameterization. (b) The RMSE from the simulation results compared to the FINO1 measurements for simulation (2), (3) and (4).*

[Figure]

*Figure R9: RMSE for three different classes of alignment (aligned 0°-60°, perpendicular 60°-120°, opposed 120-180°), for (1) the WRF only simulation (2) WRF+SWAN simulation with the roughness length parameterization of Drennan and (3) WRF+SWAN simulation with the newly proposed roughness length parameterization.*